# Cognitive Alterations in Addictive Disorders: A Translational Approach

**DOI:** 10.3390/biomedicines11071796

**Published:** 2023-06-23

**Authors:** Ani Gasparyan, Daniel Maldonado Sanchez, Francisco Navarrete, Ana Sion, Daniela Navarro, María Salud García-Gutiérrez, Gabriel Rubio Valladolid, Rosa Jurado Barba, Jorge Manzanares

**Affiliations:** 1Instituto de Neurociencias, Universidad Miguel Hernández-CSIC, Avda de Ramón y Cajal s/n, 03550 San Juan de Alicante, Spain; 2Redes de Investigación Cooperativa Orientada a Resultados en Salud (RICORS), Red de Investigación en Atención Primaria de Adicciones (RIAPAd), Instituto de Salud Carlos III, MICINN and FEDER, 28029 Madrid, Spain; 3Instituto de Investigación Sanitaria y Biomédica de Alicante (ISABIAL), 03010 Alicante, Spain; 4Instituto de Investigación i+12, Hospital Universitario 12 de Octubre, 28041 Madrid, Spain; 5Faculty of Psychology, Universidad Complutense de Madrid, 28040 Madrid, Spain; 6Department of Psychiatry, Universidad Complutense de Madrid, 28040 Madrid, Spain; 7Faculty of Health, Universidad Camilo José Cela, 28001 Madrid, Spain

**Keywords:** cognition, memory and learning, substance abuse, human studies, animal models, molecular changes

## Abstract

The cognitive decline in people with substance use disorders is well known and can be found during both the dependence and drug abstinence phases. At the clinical level, cognitive decline impairs the response to addiction treatment and increases dropout rates. It can be irreversible, even after the end of drug abuse consumption. Improving our understanding of the molecular and cellular alterations associated with cognitive decline could be essential to developing specific therapeutic strategies for its treatment. Developing animal models to simulate drug abuse-induced learning and memory alterations is critical to continue exploring this clinical situation. The main aim of this review is to summarize the most recent evidence on cognitive impairment and the associated biological markers in patients addicted to some of the most consumed drugs of abuse and in animal models simulating this clinical situation. The available information suggests the need to develop more studies to further explore the molecular alterations associated with cognitive impairment, with the ultimate goal of developing new potential therapeutic strategies.

## 1. Introduction

Cognitive alterations often appear in patients with compulsive drug consumption and are one of the main behavioral and physiological consequences of substance use disorders (SUD) [1]. Drug abuse causes structural and plastic brain changes that contribute to long-lasting behavioral alterations, including cognitive decline [2]. The development of cognitive alterations directly or indirectly increases present-day personal, family, social, and health burdens. These disturbances in adult patients are remarkably high and are even more significant in adolescents, inducing persistent brain alterations in decision-making, cognition, and emotional regulation [3,4,5,6,7].

The prevalence of cognitive alterations in patients with SUD varies considerably, from 30% to 80%, depending on the type of drugs of abuse, the age of onset, and the consumption pattern [8]. Both acute and chronic drug consumption can impair cognitive performance during post-intoxication or abstinence. Alcohol consumption has been associated with visuospatial memory loss, inhibitory performance, and impulsivity alterations. This impairment remains stable across multiple cognitive functions during the first year of abstinence, suggesting long-lasting brain alterations [9,10,11]. In alcohol use disorders (AUD), this cognitive impairment is strongly associated with Korsakoff’s syndrome, which is highly prevalent in patients with a Wernicke encephalopathy, characterized by amnesia, executive dysfunction, apathy, and affective social-cognitive impairments [12,13,14]. Long-lasting cognitive changes have also been found in opioid and cocaine users. These alterations can begin after short-term abstinence and remain for up to a year after stopping consumption [15,16,17,18,19,20,21]. In cannabis users, cognitive function impairment takes the form of alterations in emotional regulation, motor coordination, and complex executive functions such as planning, organization, and decision-making. Abstinence could be accompanied by problems in certain aspects of executive functioning, which could persist for several weeks, especially in heavy and chronic users [22,23,24,25,26]. Long-lasting cognitive disturbances are essential to predict treatment response and stable abstinence [27,28]. Indeed, several authors have reported that the cognitive impairments in patients with SUD, including attention, memory, and pre-attention, contribute to poorer treatment outcomes, decreasing treatment retention and abstinence days. These changes have been associated with less treatment adherence, lower self-efficacy, increased denial of addiction, and greater reflection impulsivity [8,27,29].

Little is known about the molecular mechanisms involved in the cognitive alterations associated with chronic and compulsive drug consumption. Each drug of abuse presents a different mechanism of action; however, all of them activate the brain reward circuit, inducing short- and long-term adaptative changes [2,30]. These changes have been incorporated recently as critical functional domains in the Addictions Neuro Clinical Assessment (ANA). Animal models have provided valuable information about the neurotransmitter system’s alterations and the usefulness of potential therapeutic interventions [31]. Different functional alterations have been found in brain regions involved in decision-making, emotional regulation, and cognitive performance, such as the prefrontal cortex (PFC), amygdala, and hippocampus (HIPP). Nevertheless, the molecular mechanisms underlying memory, recognition, or preattentional performance remain unclear.

The main aim of this review is to summarize the most recent evidence on cognitive disturbances associated with drug exposure in humans (Figure 1) and animal models and to assess the molecular mechanisms associated with the drugs to understand the biological basis of the cognitive alterations.

## 2. Materials and Methods

A comprehensive review of some of the most commonly consumed drugs, such as alcohol, cannabis, opioids, and psychostimulants, was carried out.

The review of the clinical sections describes the main findings from the last five years regarding acute and chronic drug use (assessing the effect of each substance independently and as a primary substance of abuse) and cognitive functioning. The online databases PubMed and Web of Science were searched to identify original studies, using the following search strategy: (“substance” use disorder AND cognitive deficits) OR (“substance” dependence AND cognitive deficits) OR (“substance” use disorder AND neuropsychological deficits) OR (“substance” dependence AND neuropsychological deficits). Substances tested were alcohol, cannabis, cocaine, psychostimulants, and heroin. Only studies including humans without psychiatric or medical comorbidity were reviewed.

In the preclinical sections, we selected original articles and some reviews on the effects of acute and chronic administration of different drugs of abuse (alcohol, cannabis, cocaine and other psychostimulants, and heroin) and abstinence on cognitive performance of the last 10 years. The search strategy was as follows: ((animal model AND “substance” AND cognitive deficits) OR (animal model AND “substance” AND cognitive impairment)).

## 3. Results

### 3.1. Alcohol Use Disorders

Cognitive disturbances often occur in patients with AUD, a chronic, relapsing disease associated with significant behavioral and neurological alterations. Acute alcohol exposure is associated with deficits in different cognitive aspects, especially with heavy intake. However, chronic consumption usually induces more pronounced alterations compared to acute intake. When the consumption starts in adolescence, ethanol-induced alterations generally last until adulthood, with significant cognitive impairment and brain damage [32]. These alterations could affect different aspects of cognition, such as spatial tasks, working, fear memory, and learning. This section summarizes the main findings related to cognitive alterations after different patterns of alcohol exposure in humans and animals, with a particular emphasis on acute and chronic alcohol intake.

#### 3.1.1. Clinical Studies

The neurotoxic effects of alcohol on the human brain have been extensively studied at an anatomical, physiological, and neuropsychological level. In patients with AUD, cognitive functioning and behavior are impaired [33]. These deficits are usually attributed to frontal, thalamic, and limbic regions as consequences of liver damage and poor vitamin processing, particularly thiamine [34]. Hippocampal atrophy and reduced white matter volume have been associated with long-term alcohol consumption [35]. These structural findings directly relate to altered behavior, cognitive functioning, attention span, and information speed processing [36].

Alcohol users show significant cognitive deficits in demanding tasks that require complex cognitive processing and adaptation [37]. Although there is an extensive bibliography about binge drinking, the characteristics of acute compulsive consumption differ from chronic and heavy drinking in terms of the duration of the intoxication period, the excessive concentrations of alcohol, and the accelerated alcohol metabolism resulting in impaired motor skills, blackouts and loss of consciousness [38].

The main impairments in cognitive functions reported in heavy alcohol users are attention, memory, executive functions, and social cognition [39] (Figure 1). Alcohol consumption affects visual information processing, mainly by decreasing memory and visual attention and impairing acuity and contrast sensitivity. Altered visual processing may predict adverse effects on maintaining abstinence and recovery in patients with AUD. The results of alcohol on different regions, such as the reticular formation and adjacent areas of the central nervous system, suggest an interaction with reticular regions and the cerebellum, affecting the visual cortex and its functioning [40]. Specific executive processes such as attention, working memory, inhibition, problem-solving, deduction of rules, cognitive flexibility, and impulsivity have been related to AUD, affecting the maintenance of abstinence [41]. Magnetic resonance imaging (MRI) is one of the most commonly used tools for assessing cognitive functions and brain structures [42], suggesting that patients with AUD present volume deficits in frontal, cingulate, insular, parietal, and hippocampal regions. The executive functions are directly related to the frontal cortex. Therefore, any change in this region may explain the deficit in inhibitory control, planning, and decision-making [42]. Within the components of the executive functions, working memory is one of the most studied in alcohol dependency, where impairment is associated with alcohol consumption and impulsive behavior. Khemiri and colleagues [43] conducted a randomized controlled trial for working memory training in AUD patients. The main finding was that verbal function was significantly improved after 20 sessions of working memory training over five weeks. Verbal working memory may improve in AUD patients through repeated daily training, despite continued alcohol intake during the training period.

Social cognition is significantly affected in alcohol users, who struggle with emotional regulation, socially complex situations, and recognizing negative facial stimuli, resulting in poor social skills. Cox and colleagues [44] found that the participants with AUD presented errors in recognizing emotional facial expressions such as fear or anger. These deficits in social cognition and emotional processing can contribute to family and social conflicts, resulting in interpersonal struggles as one of the leading causes of relapse [41].

Despite the considerable research on cognitive deficit in AUD patients, some authors have not found differences between control and addicted populations in neurocognitive performance in episodic memory, language, processing speed, attention, or inhibitory control [45]. However, they mention that their study’s multiple limitations (NIH Toolbox and control group) may attenuate the possible differences.

The effects of AUD tend to diminish and usually result in residual deficits when a correct recovery process is followed. Cognitive dysfunctions can improve with time and abstinence even after long-term heavy consumption [46]. The cognitive deficits described in this section could explain patients’ difficulty resisting temptation, making good choices, and selecting and planning behavioral avoidance strategies according to potential high-risk situations. Therefore, relapse rates may be high because the recovery process from AUD is more than stopping drinking—it is also rehabilitating the compromised cognitive functions [41].

#### 3.1.2. Preclinical Studies

##### Acute Alcohol Consumption

The first preclinical study on the harmful effects of acute alcohol on cognition was published in 1970, when Cox demonstrated that acute alcohol administration (2 g/kg) reduces spontaneous activity [47]. Afterward, it was noted that HIPP was a brain region most impacted by the drug and intensely involved in learning, memory, and cognition [48]. Subsequently, several authors showed that moderate doses of alcohol also impaired contextual memory and spatial variability [49,50]. Acute ethanol exposure also affects non-spatial working memory (which does not require learning or using spatial information). A more recent study demonstrated that acute ethanol exposure (only 2 days) increases astrogliosis and reduces the number of mature neurons in the HIPP’s dentate gyrus (DG), suggesting these anatomical changes are responsible, at least in part, for the behavioral and cognitive alterations found after acute ethanol intake [51].

HIPP is a critical brain region for cognitive tasks using contextual cues [52]. The effects of alcohol on contextual learning and memory have been evaluated with fear conditioning protocols [53]. In some studies, acute ethanol administered before harmful stimuli in adult rodents did not impair contextual fear conditioning [53]. However, a critical impairment was found in other experiments, where authors supported the hippocampal pCREB/CREB ratio involvement in cognitive impairment, with a higher ratio seen in animals with successful memory retention [54]. The decline in contextual fear conditioning is more pronounced than in cued fear conditioning paradigms, indicating that the brain regions involved in each situation could be differentially affected, despite the critical role of the HIPP and amygdala, because of the higher emotional implication in these experimental paradigms [53,55]. The novel object recognition (NOR) task, evaluating the sporadic recognition memory, is also impaired after pretreatment with alcohol [56]. Therefore, impaired object and spatial recognition were detected after acute early alcohol exposure with associated neuronal damage [57]. Interestingly, this adverse action occurs when alcohol is administered before the learning training session but not immediately before the evaluation [56].

##### Chronic Alcohol Consumption

Despite significant ethanol-induced cognitive impairment after acute administration, these changes are usually worse when the drug is administered chronically. Several studies have focused on evaluating chronic alcohol administration and the underlying biological mechanisms in different experimental paradigms [55]. Adolescence is a critical time point for the maturation of brain regions associated with higher-order cognitive functioning [58]. Indeed, rodents’ adolescence, which extends from the postnatal day (PND) 28 to approximately PND 35, is characterized by a higher vulnerability to external insults, modifying normal brain development [32,59]. Thus, exposure to toxic substances in this stage is associated with increased vulnerability to developing different psychiatric complications. Because of this vulnerability, animal models of early adolescent alcohol exposure have been performed to analyze long-lasting biological consequences. Fernandez and colleagues exposed early adolescent, mid-adolescent, and adult rats to a chronic intermittent ethanol (CIE) paradigm to evaluate cognitive performance at different ages. The CIE paradigm consists of intermittent alcohol exposure, using different routes of administration (depending on the study). The results revealed an impaired acquisition in non-spatial discrimination with a subsequent reversal task in CIE-exposed adult rats [60]. However, only early and mid-adolescents demonstrated impaired behavioral flexibility after CIE exposure, indicating a different age-dependent sensitivity to ethanol-induced cognitive changes. This pattern of alcohol exposure affected brain-derived neurotrophic factor (BDNF) protein levels in the PFC but not in the HIPP.

When rats were exposed to an intoxication phase, BDNF levels were reduced. Adolescent rats also showed a BDNF reduction during abstinence, whereas an increase was found in adults under the same experimental conditions [60,61]. These data suggest that adolescent alcohol exposure induces long-lasting biological alterations that persist in abstinence—effects not found after exposure during adulthood. CIE exposure also alters the activity of pyramidal neurons. Rodents exposed to this CIE model showed fewer glial progenitors in the medial PFC (mPFC), an increase in the NMDA/AMPA glutamatergic receptor’s ratio with a selective increase of the NMDA component, defining, at least in part, the cellular basis for cognitive impairments of PFC-dependent functions [62,63]. In addition, repeated CIE exposure by vapor inhalation affects behavioral flexibility in the operant chambers in adulthood. Moreover, pharmacological positive allosteric modulation of the glutamatergic mGlu5 receptors normalized the behavioral alterations acting on PFC, suggesting the involvement of the glutamatergic system in cognitive functioning [64].

Regarding the cognitive impairment in the operant set-shift paradigm (which evaluates learning and memory), CIE induced a near-complete loss of the actions of dopaminergic D2/D4 receptors on cell firing immediately, 1 and 4 weeks after the last ethanol administration [65]. These changes suggest the crucial role of dopamine receptors in the mPFC, probably by modulating the balance between excitatory and inhibitory neurotransmission. A recent study showed that chronic alcohol intake reduced excitatory thalamic inputs onto striatal cholinergic interneurons, inhibiting dopamine D1 receptors and facilitating D2-mediated responses [66]. Changes in NMDA receptors were also identified in different cortical regions after alcohol consumption [67]. Similarly, Ma and colleagues suggested the involvement of the dopaminergic system in cognitive flexibility deficiencies after chronic alcohol consumption [66]. These results point to an alteration in the excitatory and inhibitory balance in different brain regions as partially responsible for the cognitive impairment observed in alcohol-exposed rodents. The dopaminergic system could also play an essential role in this regulation. However, further studies are required to determine the real involvement of all these systems in cognitive performance.

Chronic alcohol consumption alters hippocampal neurogenesis and cell proliferation. Animal models of chronic ethanol consumption consistently identify hippocampal mass and cognitive dysfunctions as essential alcohol-related brain adaptations. Alcohol consumption induces oxidative stress and reduces neuronal stem cells and the survival rate of newborn neurons [68,69]. In addition, these new neurons present impaired spine formation with characteristic typical immature spines, indicating that alcohol could interfere with their synaptic connectivity [69]. These and other additional cellular changes can induce learning and memory deficiencies, reducing the discrimination index in the novel object recognition task and spatial and novelty memories [69,70,71]. Voltage-gate potassium channels have been identified as influencing memory-related local hippocampal circuits, partially explaining the molecular basis of the learning and memory impairments after chronic alcohol consumption [72].

Immunohistochemical studies reveal reduced neurogenesis in the DG, decreasing neuronal progenitor cell proliferation [73]. In chronic, 1-, 2-, and 4-week alcohol consumption, around 50% of cell proliferation was decreased with reduced dendritic three sizes after 4-week exposure. Alcohol consumption for three weeks impaired learning and memory in a contextual fear conditioning test and the novel object recognition test and induced increased behavioral perseveration in the Barnes maze test. In addition, 30-day prolonged ethanol consumption impaired spatial memory in the Morris water maze test, increased lipid peroxidation marker MDA levels as well as apoptotic markers caspase 3 and 9, and reduced antioxidant defenses [74,75]. Under these experimental conditions, a reduction in the DNA methylation of the *BDNF* gene was reported within the CA1 and CA3 hippocampal regions. In this study, the authors conclude that increased *BDNF*-mediated signaling is probably an adaptative process to counteract ethanol-induced cognitive deficits [74]. Seven-week ethanol exposure also induced morphological changes in the DG granular neurons and the CA1 and CA3 pyramidal neurons. Alterations in synaptic plasticity and toxicity were evaluated by quantifying NMDA glutamatergic receptor and PSD95 postsynaptic density protein. Chronic alcohol consumption and abstinence reduced DG dendritic arborizations. However, in CA1 and CA3, this effect was only observed during the exposure phase and normalized in abstinence. NMDA receptor subunits were reduced in the HIPP, supporting the involvement of the glutamatergic system in cognitive dysfunctions associated with alcohol intake [76].

##### Binge-like Ethanol Exposure

Binge-like ethanol exposure consists of administering a high amount of alcohol during short periods, periodically and intermittently, mimicking this type of alcohol intake in humans. In adolescence, binge-like ethanol exposure (2 weeks) reduced the discrimination index in the novel object recognition test [77]. Cognitive impairment also occurred in the Hebb-Williams maze analyzing visuospatial memory and learning [77]. These behavioral changes were associated with changes in CREB factor phosphorylation in the DG [77]. In addition, adolescent ethanol exposure impaired novel object recognition and increased the hippocampal Toll-like receptor 4 (*TLR4*) gene in the HIPP in adulthood, suggesting that early alcohol exposure induces long-lasting changes in the immune response [78].

A recent study revealed that adult rats exposed to a chronic adolescent alcohol protocol reduced spontaneous alternation and novel object recognition. These behavioral changes were associated with a suppressed cholinergic neuronal phenotype in the septohippocampal pathway, essential for learning and memory and altered during normal aging [79].

The nucleus accumbens (NAcc) is strongly involved in drug addiction, receiving dopaminergic projections from the ventral tegmental area in the presence of any drug of abuse [80]. Binge-like ethanol exposure impairs non-spatial memory and behavioral flexibility, inducing cholinergic alterations in the NAcc, reducing the number of cholinergic interneurons, and regulating dopaminergic neuron activity [81]. On the other hand, changes in NMDA/AMPA receptors ratio were reported under similar behavioral conditions. Of note, L-DOPA administration during the abstinence phase into the NAcc completely blocks these alterations, suggesting, as described previously, that normal dopaminergic signaling is essential during abstinence for maintaining regular neuronal activity [82].

##### Alcohol Withdrawal

Glutamatergic neurotransmission is altered after chronic drug exposure and during the withdrawal phase. In the abstinence phase, two days after the five days of ethanol administration, adolescent but not adult rats showed impaired spatial performance in the Morris water maze test [83]. Microdialysis studies revealed an increase in glutamate overflow in the CA3 region of the HIPP. Thus, higher glutamate levels in the abstinence phase appear to contribute to deficits in cognitive functions [84]. Changes in glutamatergic NMDA receptors were also identified in different cortical regions in the abstinence phase induced after 5-day alcohol administration. As described previously, NMDA glutamatergic receptors have also been altered after chronic alcohol administration, observing a significant increase in the NMDA/AMPA ratio in the PFC [63].

After chronic intermittent alcohol exposure, abstinence decreased the long-term behavioral inhibition and attentional capacity, as evaluated by the 5-Choice Serial Reaction Time Task (5-CSRTT). Withdrawal after a long period of alcohol consumption also induced impairments in object recognition memory, recognition, and conditioned taste aversion. These behavioral alterations were not found in the short-term evaluation, pointing to the long-lasting consequences of ethanol withdrawal [85]. Withdrawal after a CIE protocol causes changes in cannabinoid receptor 1 (CB1r) in the HIPP, with an initial down-regulation followed by a long-term up-regulation. At the same time, increased levels of endogenous cannabinoids were found. These results suggest that the endogenous cannabinoid system, mainly through CB1r, may be involved in the cognitive impairment induced during the abstinence phase after chronic ethanol administration [86].

The HIPP is a brain region specifically sensitive to alcohol’s adverse effects. Ethanol exposure induces structural and functional changes in this region through the glutamatergic, endocannabinoid and immune systems. These behavioral and biological disturbances were not observed in mice lacking the TLR4 receptor, suggesting the immune system’s implication in alcohol-induced disorders [87]. Therefore, the endocannabinoid, glutamatergic and immune systems could be closely involved in the effects of chronic ethanol intake.

A wide range of preclinical studies examines the effects of chronic alcohol consumption in early stages and adulthood on cognitive performance, analyzing the molecular mechanisms involved (Table 1). The brain regions related to cognitive impairment after alcohol intake are the PFC, HIPP, and NAcc. These regions control executive functions, cognition, memory, and learning, with significant changes in different components of the cell excitatory and inhibitory pathways, the dopaminergic, cholinergic, and immune systems, and structural changes in cell morphology. Despite these results, the exact mechanism by which alcohol impairs cognition after its chronic use remains unclear, suggesting the need to further explore this field and the molecular mechanisms involved.

### 3.2. Cannabis Use Disorders

#### 3.2.1. Clinical Studies

The scientific literature has broadly investigated cognitive disruptions induced by cannabis use [88,89,90,91,92,93,94,95]. The acute effects of cannabis and different patterns of consumption during active use and abstinence have been evaluated for possible psychiatric (e.g., proneness to psychotic episodes or schizophrenia) [93] and cognitive effects (in memory, speed processing, attention, or executive functions) [89,90,91,93]. Nonetheless, the need to revise the methodology, homogeneity of studies and criteria for participant selection, clinical characteristics or follow-up measures is relevant for identifying maintained alterations produced by cannabis in high-order cognitive functioning [90,92]. This section summarizes the main recent findings of cannabis use effects in youth and adults according to cognitive domain.

Individuals with cannabis use disorders (CUD) who attend treatment facilities show poor scores in visuospatial processing tests in cross-sectional [88,96] and longitudinal studies, especially adolescents with heavy cannabis use [91,97]. In some studies, the rate of patients abstinent from other drug or marijuana use was low (between 20% and 30%), and it implied a few days or weeks without consumption, which means that we cannot entirely rule out the lasting effect of active cannabis and/or other drug use [96,98]. While visuospatial abilities did not seem affected in one sample of young cannabis users that were abstinent for three weeks [99], other studies have shown that visuospatial abilities might be altered by heavy and regular cannabis use after short to medium periods of abstinence (weeks or months), though they might not persist in time [91]. Psychomotor function and speed are impaired in regular [100] and chronic cannabis users in finger tapping tasks [89,101]. This effect is maintained within hours of abstinence (residual effects) [91,102], and a deficit in psychomotor speed is related to past cannabis use in young people abstinent for three weeks [99].

Attentional processes have been highly studied in cannabis use, and they show alterations within short periods of abstinence or active use [95,103]. In a few recent studies, speeded attention does not seem impaired by active daily cannabis use in young adults [96,104]. A recent systematic review found similar performances between chronic cannabis users and non-users on a visual attentional task [89]. Conversely, sustained attention seems to be affected in a cross-sectional CUD study using the Continuous Performance Task [96]. However, a survey of a large sample of the general population found no apparent effects on sustained attention, though only a small proportion regularly and/or chronically used cannabis [105]. Other recent research has found slower reaction times and, thus, slower processing during sustained attention tasks in regular cannabis users [106] and long-term users with hours of abstinence [107]. Additionally, complex attention (e.g., involving language or working memory tasks), but not attentional control, is another process that might be affected, even after weeks or months after ceasing cannabis consumption in heavy cannabis users [91,93,97].

Verbal learning and memory deficits have been consistently found in cannabis users [89,90,91,93,103,104,107,108], and seem to be impaired with chronic use in longitudinal studies [104]. Specific processes such as verbal encoding, recall and recognition appear altered by cannabis use, though not all studies replicated this result [96].

A memory process that has less consistent evidence of alterations by cannabis use is working memory [93,108]. For instance, while some studies in young people with daily cannabis use find working memory deficits [93,95], others do not replicate this result in the general population [105]. These results should be interpreted carefully since not all studies report abstinence or involve only days without cannabis use. Hence, most studies cannot exclude the recent toxic or present effects of cannabis, even in chronic users. In this way, when abstinence is considered, after 24 h of no cannabis use, chronic users perform similarly to controls in the verbal memory task, though learning deficits might persist [89,109]. This latter effect in verbal learning was found with hours of abstinence in adolescent-onset cannabis users [91,110], in long-term users [107], and in adults with CUD [96]. In this way, cannabis use seems to have consistent short-term and long-term effects on learning and memory processes, though there might be limited evidence for its lasting effects after long periods of abstinence [93]. Some inconsistencies across the studies regarding memory process impairment or the lasting effects of cannabis use might be explained by other variables. In this line, increasing years of cannabis use, age of onset, frequency of use, etc., are related to poorer memory task performance in abstinent adolescents [90] and adults [93].

Many components of executive functioning, such as inhibition, planning, decision-making, and cognitive flexibility have been studied in cannabis users. The chronic effect of cannabis use on inhibition may not be precise enough [93,111] in either motor [96,107] or cognitive inhibition [105] tasks. Cannabis use might affect error awareness [89]. Nonetheless, the age of onset might be relevant to the disparities in the results and the clinical severity. One narrative review found that relapse impairment and treatment abandonment might be related to higher impulsivity [108]. Persistent deficits in planning processes, such as Tower of London performances, were reported during active chronic cannabis use in young people [104]. The authors interpreted this finding as a possible reflection of motivational difficulties or amotivation since the items used in the Tower of London had relatively easy solutions. Amotivation might be characteristic of cannabis use effects, possibly affecting daily life, manifested as a lack of engagement in certain “less demanding” activities. One study found that effort in cognitive tasks might mediate managing cognitive demands, such as memory [112]. They also seem to improve with long periods of abstinence (3 years) [91].

Motivated decision-making, assessed by the Iowa Gambling task, is impaired in young daily cannabis users, though the performance might improve over time [91,104]. This could be partially explained by reducing cannabis use or abstinence and the protective factors of ongoing neurodevelopment, which is not over until about 22–23 years of age. Chronic use of cannabis in adults results in decision-making deficits in behavioral paradigms evaluation after short periods of abstinence (8 h to 1 day) and longer ones (25 days), as found by systematic reviews [89]. These findings might reflect a reduced sensitivity to the negative consequences of behavior in the cost-benefit balance of reward processing. Nonetheless, recent reviews show inconsistencies regarding cannabis’s long-term effects on motivated decision-making [93], where current users did not differ from controls and in a delay discounting task, abstinent users had a similar performance to non-users [90,93].

Although many studies have evaluated executive functions in cannabis use, cognitive flexibility has not been found to be impaired [88,105]. However, a recent systematic review found insufficient evidence of its deficit in chronic cannabis users [92].

##### Factors Associated with Regular and/or Chronic Cannabis Use

Considering the above, adolescents and adult cannabis users might show deficits in several crucial domains for daily life functioning: memory and learning processing, attention and processing speed, and some executive functions such as motivated decision-making [90,91,95,103]. Some issues or factors may influence these effects: age of onset consumption [89,91,104], frequency of cannabis use [91,113], consumption intensity [89], THC concentrations [89], and total duration of use. Nonetheless, other important factors must be considered when assessing the effects of cannabis use on cognition that can act as possible confounders: sociodemographic variables, e.g., age, sex [90,100], and years of education, and clinical variables such as other drug use (tobacco, alcohol, stimulants, etc.) [91] and mental disorders (criteria or traits for personality disorders, vulnerability to psychiatric conditions or comorbidities). Indeed, Becker and colleagues (2018) [104] studied the neuropsychological performance of young, non-treatment-seeking users of cannabis with no comorbid psychopathology. They controlled for sex, age, IQ, baseline performances and time between assessments. Cannabis users initiated consumption around age 17 and were 18 to 19 years old at baseline, when they showed alterations in general memory and executive functioning. Two years later, they maintained similar cannabis use patterns (daily and/or heavy use), and deficits were persistent in working memory, verbal memory, and planning. These correlated with age onset, possibly reflecting a worse cognitive performance with earlier initiation of cannabis consumption. Although some cognitive alterations persisted at follow-up, they did not seem to worsen over time. Thus, there are no apparent signs of cognitive decline. Similar findings were found in a longitudinal co-twin study, where no cognitive deficits were observed with cannabis use. However, authors reported that early ages of consumption (17 years) were associated with worse executive functioning at later ages (23) [111].

Assessments of last consumption and abstinence monitoring are crucial for evaluating the lasting effects of cannabis on cognition and behavior [90,104]. Moreover, without knowing the timing of last consumption, interpreting data can be difficult [92], precluding a clear distinction between critical aspects such as continuous intoxication effects of active or recent consumption and toxic-free lasting effects on cognition during abstinence periods. A summary of the active or current effects of cannabis use and abstinence (when considered or mentioned in the reviewed studies) is presented below.

##### Ongoing Active Effects of Regular and/or Chronic Cannabis Consumption

Although recent use of cannabis could be related to some neuropsychological deficits [94,105], the lasting ones are less clear. Some alterations in memory and learning persist in the first hours of non-consumption of cannabis [107], and in some studies, even after days [94]. However, no clear evidence of persistent cognitive alterations is revealed after weeks and months of abstinence [90]. A recent systematic review and meta-analysis of the chronic effects of cannabis excluded participants who were not abstinent or presented other clinical disorders (including other substance use disorders) and reported inconsistent findings in cognitive processes [92]. Some reviews have concluded that long-term effects might only be present in heavy, chronic cannabis users with an early age of onset [94]. Recent research reveals worse self-reported cognitive functioning in older active or recent cannabis users, especially those with CUD [96,113]. Active users of cannabis also reported worse mental health. They met the criteria for mental health disorders and showed an earlier age at onset consumption, which may be an essential issue when interpreting the cognitive results. For instance, almost 30% of participants from the study by Benitez and colleagues [113] had a concurrent alcohol use disorder. Hence, a straightforward evaluation and description of clinical characteristics is needed in further research.

##### Post-Detoxification Effects Related to Regular and/or Chronic Cannabis Consumption

Studies evaluating the post-detoxification effects of regular or chronic cannabis consumption show that some cognitive difficulties may persist for short- to mid-term periods of abstinence [91,92,96]. However, in the long-term, full cognitive recovery can be expected. For instance, Bruijnen and colleagues (2019) did not find evident cognitive impairments in a cognitive screening assessment of chronic cannabis users who attended treatment care centers. However, their results may not be precise enough due to possible confounding effects of the duration of abstinence (only 16 out of 123 participants had more than 7 days of abstinence) and polysubstance use (66 out of 123 participants had a secondary consumption of other drugs) [30]. A recent systematic review investigated residual (with less than 14 days of abstinence) and sustained (2–4 weeks to months or even years) cannabis effects on cognitive processes. They found variable results, possibly about other substance use effects or variations of abstinence across the studies. Consistent findings are shown in complex attention, visuospatial performance, and memory [91].

Overall, cannabis’s effects on cognitive function seem evident in the short term, especially in regular and daily users, in functions such as attention, information processing speed, visuospatial performance and learning and memory processes. These might improve over time if cannabis consumption ceases and with long periods of abstinence (after several months or years). Nevertheless, clinical variables related to psychopathology and drug use, such as years of consumption, severity, etc., might play an essential role in the persistence of cognitive deficits. There is an evident need for research controlling for these variables and evaluating the long-term effects of cannabis and specifically for longitudinal studies assessing several periods of abstinence.

#### 3.2.2. Preclinical Studies

The worrying increase in cannabis use, accompanied by a marked reduction in risk perception, particularly among the adolescent population, has prompted numerous studies in animal models that can provide valuable information on the associated harms and underlying neurobiological mechanisms involved from a translational perspective. The following sections compile the evidence on the effects of acute or chronic exposure to Δ^9^-THC (Table 2) in various species and animal models on cognitive, molecular, and functional aspects.

##### Acute Exposure to Δ^9^-THC

Some of the first studies evaluating the acute effects of Δ^9^-THC were carried out in primates, providing relevant information about the complex alterations in cognitive behavior in male rhesus macaques. Taffe and colleagues showed that Δ^9^-THC impaired working and visuospatial associative memory and reduced incremental learning in progressive ratio and bimanual motor skill tasks in adult male macaques [114]. Likewise, similar dosages also altered spatial working memory in the delayed match-to-sample task in adolescent-exposed primates [115]. Wright and colleagues performed an exhaustive battery of behavioral tests to characterize further the acute Δ^9^-THC-induced cognitive disturbances. The results revealed poorer motor skill performance and alterations in reinforcement and visuospatial associative learning and spatial working memory [116].

The vast majority of studies on Δ^9^-THC and cognitive performance under acute or repeated systemic administration patterns have been carried out in rodents. Acute exposure to Δ^9^-THC induces significant alterations in effort-related decision-making [117], spatial learning, and working memory in several behavioral paradigms such as the eight-arm radial maze [118], the Y-maze [119], the Barnes maze [120], and the delayed response task [121]. In addition, impairments in recognition memory were identified in the novel object recognition (NOR) task [122,123,124,125].

The improvement of knowledge and understanding of the molecular and functional changes resulting from Δ^9^-THC exposure is essential to identify targets for developing new therapeutic strategies to treat the damaging psychotropic effects of cannabis. Experimental models of acute exposure have contributed to characterizing the endocannabinoid system’s role. For instance, the protein density of CB1r in the PFC is proportionally correlated with Δ^9^-THC-induced impairments in decision-making in a cognitive effort task [117]. Moreover, there is growing interest in the functional role of CB1r form heteromers with other receptors in response to Δ^9^-THC administration. Viñals and colleagues proposed that CB1r heteromers and the serotonin 2A receptor (5HT2Ar) are expressed in brain regions involved in memory impairment and play a crucial role in the dissociation of detrimental (cognitive deficits) and beneficial (antinociception) properties of Δ^9^-THC [122]. The presence of heteromers between CB1r and adenosine 2A receptor (A2Ar) was first described by Aso and colleagues at the presynaptic level in the CA1 sub-region of the HIPP. Importantly, these CB1r-A2Ar heteromers may have a pivotal role in the protective effects of CBD on Δ^9^-THC-induced cognitive disturbances in declarative and spatial memory [125].

The acute administration of Δ^9^-THC did not cause significant changes in synaptic transmission in hippocampal slices from adult male CD1 mice exposed to the novel object recognition test (NOR) [123]. However, similar experimental conditions with the same Δ^9^-THC dosages induced a transient increase in hippocampal protein kinase C (PKC)/neurogranin signaling pathway, correlated with alterations in short-term object recognition memory [124].

One aspect that has been the focus of research on the consequences of Δ^9^-THC exposure in animal models is the distinction between the effects induced in adolescence and those produced in adulthood [126]. Indeed, a study performed in female CD1 mice evaluated differential Δ^9^-THC-induced disturbances in the brain lipidome and transcriptome at different age stages, concretely post-natal day (PND) 35, PND 50, and adulthood (≈4 months old). The most relevant changes involving endocannabinoids (eCBs) and structural analogs occurred in adulthood and revealed a significant generalized decrease [127]. Likewise, transcriptomics analysis in the HIPP showed that the number of transcripts whose expression changed significantly was higher during adulthood (n = 158) than in adolescence (n = 58). However, the regulation of 31 genes overlapped between stages, suggesting a conserved response to Δ^9^-THC acute administration at the transcriptomic level [127]. A study in adult male C57BL/6J mice administered with an acute amnesic dose of Δ^9^-THC (10 mg/kg, i.p.) pointed to a significant impairment of the synaptosomal proteome in the HIPP. This result reveals a significant alteration of the proteasome system that could be involved in aberrant plasticity at synapses [128].

Apart from the studies evaluating the systemic behavioral and molecular brain effects of Δ^9^-THC, some authors investigated the actions derived from intracerebral administration in specific brain nuclei in adult Sprague-Dawley rats. Acute intra-PFC microinfusion did not change cognitive flexibility or spatial working memory [129]. Nevertheless, acute Δ^9^-THC microinfusion in the ventral HIPP (vHIPP) [130] and the posterior shell region of the NAcc (NAcc-Sh) [131] notably increased the freezing time to contextual or conditioned stimuli, increasing the recall of the traumatic situation. Furthermore, intra-NAcc-Sh administration of Δ^9^-THC impaired sensorimotor gating in the pre-pulse inhibition task [131], showing pre-attentional deficits in cognitive performance [132,133]. These alterations include profound changes in relevant intracellular signaling pathways such as phosphatidylinositol 3-kinase (PI3K)/AKT, mitogen-activated protein kinase (MAPK)/extracellular signal-regulated kinase (ERK), and glycogen-synthase-kinase-3 (GSK-3)-β-catenin, all involved in cell maintenance, survival, and growth in specific brain regions [130,131]. In addition, intra-vHIPP microinfusion of Δ^9^-THC impaired the neuronal activity of the ventral tegmental area (VTA) dopaminergic and GABAergic neurons [130], showing the potential involvement of these systems in the cognitive impairment induced by Δ^9^-THC administration.

##### Chronic Exposure to Δ^9^-THC

The studies of acute Δ^9^-THC administration on impaired cognition are limited, since they do not reproduce the type of cannabis consumption seen from a clinical perspective. Thus, animal models evaluating the effects of repeated exposure receive more attention due to their higher translational potential.

John and colleagues evaluated the differences between the consequences of acute and chronic Δ^9^-THC administration in adult male rhesus macaques [134], showing that impairments in working memory were independent of the type of administration, whereas stimulus discrimination and extradimensional set-shifting performance were only affected in acutely treated animals. The D2/D3r striatal availability was no different between chronically Δ^9^-THC-treated and control primates, a finding consistent with those previously obtained in patients under baseline conditions [135,136,137].

Prolonged exposure to low Δ^9^-THC doses (15–240 µg/kg, i.v.) in adolescent male rhesus macaques induced spatial working memory-related cognitive disturbances [138] and spatial delayed repose (SDR) [139] paradigms. Higher repeated Δ^9^-THC doses (0.1–1 mg/kg, i.m.) in adolescent Squirrel monkeys affected discrimination learning without modifying cognitive flexibility [140].

Several studies evaluating the effects of chronic exposure to Δ^9^-THC have been carried out in adolescent animals to assess the long-term consequences during adulthood. Scarce literature is available on the actions of Δ^9^-THC administered only in adult rodents. A reduction in the percentage of prepulse inhibition (PPI) was induced by Δ^9^-THC (1 mg/kg, i.p., 21 days) in adult male Sprague-Dawley rats [141]. Furthermore, Chen and colleagues reported that chronic exposure (10 mg/kg, i.p., 7 days) impaired spatial and fear-related memory and hippocampal dysfunction, decreased long-term potentiation (LTP), induced a reduction of dendritic spines and postsynaptic density, and down-regulated glutamatergic neurotransmission [142]. Repeated Δ^9^-THC administration (10 mg/kg, i.p. 14 days) disrupted the balance of corticolimbic glutamatergic input to the NAcc [143]. Thus, even during the adult stage, Δ^9^-THC produced critical changes in functionality and synaptic connectivity in brain regions involved in memory regulation.

The numerous rodent studies exposing animals to Δ^9^-THC during their adolescence and evaluating behavioral, molecular, or functional deficits during adulthood could be classified by administration pattern. Some used a fixed dose, others a variable frequency of administration, and others still a design in which the Δ^9^-THC dose progressively increased (a more translational approach).

Chronic Δ^9^-THC administration (10 mg/kg, i.p., 30 days) did not affect PPI in adolescent CD1 mice, but in combination with a 5HT2Ar receptor agonist there was an exacerbation in the sensorimotor gating disruption [144]. On the other hand, adolescent C57BL/6J mice exposed to Δ^9^-THC (5 mg/kg, i.p., PND 30–45) had impaired object and social recognition memory, together with profound functional and connectivity alterations in the PFC that persisted in adulthood [145]. Under similar exposure conditions, Le and colleagues showed alterations in episodic and spatial memory, particularly in female Long-Evans rats and C57BL/6J mice, which were associated with impaired mechanisms of LTP synaptic plasticity in the HIPP [146]. In addition, impaired dendritic arborization, spine density, and developmental trajectory of layer III pyramidal neurons [147] were found in adolescent male Long-Evans rats, which may be closely related to potential cognitive disturbances induced by Δ^9^-THC administration. These neuronal morphological alterations consist of neuroplastic changes intensely involved in cognition and learning. Therefore, any alteration in neuronal morphology and connectivity could alter the average cognitive performance.

One of the few studies employing an intravenous operant Δ^9^-THC self-administration paradigm improved working memory performance in adolescent Sprague-Dawley rats in the delayed match-to-sample (DMTS) task. This enhancing effect reduced CB1r, GABAergic and glutamatergic receptor function, particularly in the PrL [148]. A similar improvement on spatial and working memory was found after chronic high-dose administration of Δ^9^-THC (60 mg/kg, i.p., 28 days) in the Morris water maze [149]. These results are consistent with those obtained by Hudson and colleagues regarding the positive effects of Δ^9^-THC on cognition [131]. However, despite these results being inconsistent with the available literature, the molecular mechanisms involved in these possible positive actions remain unclear. Therefore, further studies are required to explore these discrepant findings on the actions of Δ^9^-THC on cognition.

Mice deficient for the polysialyltransferase ST8SIA2 (ST8SIA2−/−), involved in the glycosylation of the neural cell adhesion molecule (NCAM), but not the corresponding controls (WT mice), were characterized by poorer spatial reference memory and lower learning index in the hole board paradigm three months after juvenile chronic exposure to Δ^9^-THC (7 mg/kg, i.p., every other day for 21 days) [150]. Moreover, another mouse genotype was employed to evaluate the role of the Dominant-Negative Disrupted-In-Schizophrenia-1 (DN-DISC1) gene. Chronic Δ^9^-THC administration induced alterations in object and spatial recognition memory involving BDNF expression in the HIPP [151] and the activation of NF-kB-COX-2 signaling in astrocytes only in homozygous DN-DISC1 mice [152]. These studies, employing genetically engineered mouse models, provide relevant data about the critical interaction between genetic and environmental (Δ^9^-THC exposure) factors during adolescence and the long-lasting adverse effects on cognition in the adult stage.

Some studies have explored the effects of chronic Δ^9^-THC vapor exposure in adolescent rats under different experimental conditions (e.g., Δ^9^-THC dosage, time of exposure during each session, total duration of exposure, or combination with ethanol exposure). Prolonged exposure to smoke (5.6% Δ^9^-THC, PND 29–49) in Long-Evans adolescent rats only impaired working memory, without affecting motivational learning, decision making, or cognitive flexibility during adulthood [153]. In Sprague-Dawley adolescent rats, chronic vapor exposure every other day did not induce recognition or aversive memory changes in adult animals. However, the combination of Δ^9^-THC with alcohol during the adolescent period (PND 28–42) significantly reduced the discrimination index of NOR [154]. Likewise, Wistar rats exposed to Δ^9^-THC in inhalation chambers, alone or combined with a protocol of CIE exposure, increased freezing time in the fear conditioning paradigm, probably by altering fear sensitivity [155]. Thus, it could be hypothesized that the co-exposure to Δ^9^-THC and alcohol might heighten long-term cognitive disturbances and alter fear stimuli processing. However, further studies are required to clarify this aspect.

Most recently, rodent studies employed a progressive Δ^9^-THC exposure protocol during adolescence to elucidate behavioral and neurobiological alterations in the adult stage. This protocol usually consists of twice-daily administration, first of a low Δ^9^-THC dose (2.5 mg/kg, i.p.) from PND 35 to PND 37, followed by 5 mg/kg (PND 38–41) and finally 10 mg/kg (PND 42–45) [156,157,158,159,160,161,162,163], inducing long-term cognitive and neurobiological disturbances.

The NOR test has been the most widely used paradigm to assess the effects of prolonged and progressive exposure to THC during adolescence. Considering both classic and spatial versions, progressive adolescent exposure to Δ^9^-THC impaired recognition and spatial memory in male [158,159,164,165] and female [156,161,162,166] rats. Nevertheless, the NOR showed no significant differences between adolescent-treated male and female Long-Evans rats [163]. However, the spatial version of the NOR did show an improvement in spatial memory [165].

PPI deficits were also induced in male rats [159,160], although they were absent when employing significantly lower Δ^9^-THC doses (0.3, 1, 3 mg/kg) under the previously described administration protocol [164] or in Wistar rats [166]. Other authors found long-term cognitive impairments with reduced social recognition memory [159] or spatial memory in the Morris water maze [164] and the water T-maze [165].

In rodents exposed to a model of repeated and progressive exposure to Δ^9^-THC during the adolescent stage (Table 2), the PFC and the HIPP represent two crucial regions modulating cognition.

Neuroinflammatory and oxidative processes have been described in the PFC, with increased levels of ionized calcium-binding adapter molecule 1 (Iba1), tumor necrosis factor-alpha (TNFα) and inducible nitric oxide synthetase (iNOS), cyclooxygenase 2 (COX-2), and CD11b antigen [156,158,161]. Furthermore, while CB1r is usually down-regulated [156,157,161], CB2r has been up-regulated [156], the latter finding being consistent with the neuroprotective effects that have been described for this cannabinoid receptor [167]. Adolescent exposure to Δ^9^-THC in rodents revealed profound functional disturbances in the PFC in intracellular signaling [159], excitatory/inhibitory balance [161], and synaptic plasticity [157,162,164,166]. There is increasing evidence of the imprint that early exposure to cannabis induces in the maturation and development processes of the PFC, a fact that is directly associated with later cognitive impairment [168,169,170].

In the HIPP, severe alterations in synaptic plasticity and neurogenesis play a crucial role in regulating learning and memory processes. These disturbances account for the evident cognitive deficits in the non-spatial memory (NOR) and pre-attentional deficits evaluations (PPI). Increased levels of neuroinflammatory factors such as TNFα, iNOS, or COX-2 were described together with the up-regulation of glutamate receptors (GluA1, GluA2, and GluN2B) [158]. Notably, reduced doublecortin and SRY (sex-determining region Y)-box 2 positive cells (DCX+ and SOX2, respectively), decreased dendritic length of DCX+ cells, and impaired LTP in the DG of the HIPP were discovered in adolescent female Sprague-Dawley rats [157,164].

**Table 2 biomedicines-11-01796-t002:** Summary of the findings related to Δ^9^-THC-induced cognitive and biological changes in animal models.

ACUTE Δ^9^-THC					
Pattern, Doses and Route of Administration	Strain and Stage of Administration	Test	Behavioral Alterations	Biological Alterations	References
Acute, (0.1–0.3 mg/kg, i.m.) 45 min before behavioral testing	Adult male rhesus macaques	Self-ordered spatial search Visuo-spatial paired associates learning PR schedule of reinforcement & bimanual motor skill	↓ Working memory ↓ Visuo-Spatial associative memory↓ Incremental learning	-	[114]
Acute (30–240 μg/kg, i.v.) 30 min before behavioral testing		Spatial delayed response Delayed match-to-sample	=Object Working Memory ↓ Spatial Working Memory		[115]
Acute, (0.2, 0.5 mg/kg, i.m.) 30 min before behavioral testing		Bimanual motor skill Visuo-spatial paired associates learning Self-ordered spatial search PR schedule of reinforcement Rotating turntable task	↓ Behavioral performance ↓ Visuospatial associative learning and memory ↓ Spatial working memory ↓ Number of reinforcements ↓ Threshold retrieval speed		[116]
Acute (6 mg/kg, i.p.) 60 min before behavioral testing	Adult male Wistar rats	Eight-arm radial maze	↓ Spatial memory	-	[118]
Acute, (1.25–25 mg/kg, i.p.) 50 min before behavioral testing	Adult male CD1 mice	Y-maze	↓ Spatial learning and memory performance	-	[119]
Acute, (0.1–3 mg/kg, i.p.) 15 min after training		Novel object recognition	↓ Short- and long-term recognition memory (discrimination index) ↓ Total object exploration	= Synaptic excitatory transmission (hippocampal slices) = K^+^-evoked glutamate and GABA release (hippocampal slices)	[123]
Acute, (0.1, 0.3, 1, 3 mg/kg, i.p.) 20 min after training			↓ Recognition of short- and long-term memory (discrimination index)	↑ Phosphorylated substrates of PKC in the HIPP ↑ Phosphorylated neurogranin in the HIPP	[124]
Acute, (0–10 mg/kg, i.p.) 15 min before behavioral testing	Adult male C57BL/6J mice	Conditional discrimination Barnes maze	=Discrimination ratio ↓ Spatial memory (THC-treated mice)	-	[120]
Acute (1 and 3 mg/kg, i.p.) immediately after training		Novel object recognition	↓ Recognition memory (discrimination index)	Presence of A2Ar and CB1r heteromers at the presynaptic level in CA1 neurons in the HIPP	[125]
Acute (3 and 10 mg/kg, i.p.) immediately after training			↓ Recognition memory (discrimination index) in WT mice	CB1r-5HT2Ar heteromers are involved in the amnesic effect of Δ^9^-THC (cortex, striatum, NAcc, HIPP)	[122]
Acute, (0, 0.3, 1, 2, 3 mg/kg, i.p.) 30 min before behavioral testing	Adult male Long-Evans rats	Rat cognitive effort task	Impairment of decision-making involving cognitive effort costs	CB1r density in the mPFC correlated with Δ^9^-THC-induced choice impairments	[117]
Acute intra-PFC, (10, 50, 100 or 500 ng/0.5 µL)	Adult male Sprague-Dawley rats	Attentional set-shifting Y-maze	=Cognitive flexibility =Spatial working memory	-	[129]
Acute intra-vHipp, (10, 100 ng/0.5 µL)		Context-dependent and context-independent fear conditioning	↑ Freezing time (to context and conditioned stimulus)	↑ pERK1-2 protein expression in the vHipp ↑ VTA DA frequency and bursting rates ↓ VTA putative GABAergic neuronal activity	[130]
Acute intra NAcc-Sh, (100 ng/0.5 μL)		Olfactory fear conditioning Prepulse inhibition Novel object recognition	↑ Freezing time (posterior NAcc-Sh, sub-threshold fear memory 0.4 mA footshock) ↓ Freezing time (anterior NAcc-Sh, supra-threshold fear memory 0.8 mA footshock) ↓ PPI and ↑ PPF (posterior NAcc-Sh) ↓ Discrimination index (posterior NAcc-Sh)	↓ pAktSer473 and ratio pAkt:tAktSer473 (anterior NAcc-Sh) ↓ pmTOR and ratio pmTOR:tmTOR (anterior NAcc-Sh) ↓ pGSK3a and pGSK3a:tGSK3a (posterior NAcc-Sh) ↑ β-catenin (posterior NAcc-Sh)	[131]
Acute smoke exposure (0, 1, 3, 5 cigarettes with 5.6% of Δ^9^-THC, inh.) 15 min before behavioral testing Acute, (0, 0.3, 1, 3 mg/kg, i.p.) 10 min before behavioral testing	Adult male and female Long-Evans rats	Delayed response working memory task	↑ Working memory accuracy (females) ↓ Working memory accuracy (males and females)	-	[121]
Acute, (3 mg/kg, i.p.)	Adolescent and adult female CD1 mice	-	-	Significant changes in the mouse brain lipidome (PND 35, PND 50 and adulthood) Significant changes in the brain levels of targeted lipids, including eCBs (PND 35, PND 50 and adulthood) Significant changes in the HIPP transcriptome (PND 35, PND 50 and adulthood)	[127]
Acute, (10 mg/kg, i.p.)	Adult male C57BL/6J mice			Impairment of the hippocampal synaptosomal proteome (metabolic pathways and proteasome system)	[128]
**CHRONIC Δ^9^-THC**					
**Pattern, Doses and Route of Administration**	**Strain and Stage of Administration**	**Test**	**Behavioral Alterations**	**Biological Alterations**	**References**
Acute (0.01–0.56 mg/kg, i.v.) and chronic (1.0–2.0 mg/kg, i.v., 12 weeks)	Adult male rhesus macaques	Delayed match-to-sample Stimulus discrimination Reversal learning Attentional set-shifting	↓ Working memory (acute and chronic admin.) ↓ Compound discrimination (acute admin.) No significant effects ↓ Extradimensional set-shifting performance (acute admin.)	=D2/D3r striatal availability	[134]
Chronic (15–240 µg/kg, i.v., 5 days per week, 6 months)	Adolescent male rhesus macaques	Spatial memory task Object memory task	Impaired improvements in accuracy on the spatial working memory No significant effects	-	[138]
Chronic (15–240 µg/kg, 12 months, i.v.)		Spatial delayed response	Impairment of reinforcement-related learning processes required for improved performance on spatial working memory		[139]
Chronic, (week 1: 0.1 mg/kg; week 2: 0.3 mg/kg; week 3 and during 4 months: 1 mg/kg; i.m.)	Adolescent male Squirrel monkeys	Repeated acquisition Discrimination reversal	↓ Discrimination learning =Cognitive flexibility		[140]
Chronic (1 mg/kg, i.p., once daily for 21 days)	Adult male Sprague-Dawley rats	Prepulse inhibition	↓ % of PPI	-	[141]
Chronic (10 mg/kg, i.p., once daily for 7 days)	Adult male C57BL/6J mice	Morris water maze Fear conditioning	↓ Spatial memory ↓ Fear memory	↑ COX-2 expression (CB1r-dependent) ↓Long-term potentiation at hippocampal CA3-CA1 synapses ↓ Dendritic spines and postsynaptic density ↓ GluA1, GluN2A, GluN2B protein expression in the hippocampal CA1 area	[142]
Chronic (5 mg/kg, 14 days, i.p.)	Adult male Long-Evans rats	-	-	Disruption of the balance of corticolimbic glutamatergic input to the NAcc (mostly prevented by CB1r antagonism)	[143]
Chronic (10 mg/kg, i.p., 30 days)	Adolescent male CD1 mice	Prepulse inhibition	↑ PPI disruption induced by the activation of 5HT2Ar	=5HT2Ar protein density ↑ 5HT2Ar signaling through inhibitory G-proteins Involvement of Akt/mTOR intracellular signaling pathway	[144]
Chronic, (5 mg/kg, PND 30–45, i.p.)	Adolescent male C57BL/6J mice	Novel object recognition Social discrimination Task	↓ Recognition memory (discrimination index) ↓ Social recognition memory (discrimination index)	Long-lasting activation of mTOR in the PFC that persisted in adulthood Impaired excitatory and inhibitory transmission in the PFC Impaired intrinsic properties of layer V pyramidal neurons Impaired LTD at PFC layer I/V synapses	[145]
Chronic (1.5 mg/kg, every 3rd day for a total of 8 Injections, i.p.)	Adolescent male Long-Evans rats	-	-	↑ Plasma corticosterone Impaired dendritic arborization, spine density, and developmental trajectory of layer III PrL pyramidal neurons Impairment of normal developmental trajectory of the PrL pyramidal transcriptome	[147]
Chronic (5 mg/kg, 14 days PND 30–43, i.p.)	Adolescent male and female C57BL/6J mice Adolescent male and female Long-Evans rats	Serial “what” task and two-odor discrimination Novel object recognition (spatial version)	Impaired episodic memory (↓ discrimination) ↓ Discrimination index (female rats and mice)	Impaired mechanisms of enduring, memory-related synaptic plasticity (LTP) within HIPP (female rats and mice)	[146]
Self-admin. (3–100 µg/kg, i.v.)	Adolescent male and female Sprague-Dawley rats	Delayed match-to-sample	↑ Working memory performance in male rats	↓ CB1r protein expression in the PrL, IL and VTA ↓ GABAAR1α protein expression in the PrL and DH ↓ GABABr2 protein expression in the PrL ↓ GluR2/3 protein expression in the PrL	[148]
Chronic (60 mg/kg, 28 days, i.p.)	Adolescent and adult female Wistar rats	Morris water maze	↑ Spatial and working memory	=BDNF protein levels	[149]
Chronic (7 mg/kg, i.p., every other day for 21 days)	Adolescent male St8sia2−/− and St8sia2+/+ mice (C57BL/6N background)	Hole board paradigm	↓ Spatial reference memory in St8sia2−/− ↓ Learning index in St8sia2−/−	↑ PolySia levels in the HIPP of St8sia2−/− ↓ PolySia-free NCAM-180 in the HIPP of St8sia2−/− ↑ PolySia in the molecular layer of the HIPP of St8sia2−/−	[150]
Chronic treatment (10 mg/kg, PND 42–51, i.p.)	Adolescent male and female DN-DISC1 and WT mice (C57BL/6J background)	Novel object recognition	↓ Recognition memory (discrimination index) in DN-DISC1 mice	↑ BDNF in the HIPP in WT mice	[151]
Chronic (8 mg/kg, 3 weeks from PND 30, s.c.)		Y maze Novel object recognition Novel place recognition test	↓ Spatial recognition memory in DN-DISC1 male and female mice ↓ Recognition memory (discrimination index) in DN-DISC1 male mice↓ Preference for the novel place of one of two identical objects in DN-DISC1 male mice	↑ Activation of the NF-kB-COX-2 pathway in astrocytes ↓ Immunoreactivity of parvalbumin-positive pre-synaptic inhibitory boutons around pyramidal neurons of the hippocampal CA3 area	[152]
Chronic smoke exposure (5 cigarettes/day from PND 29 to 49, 5.6% of Δ^9^-THC, inh.)	Adolescent male Long-Evans rats	PR Delayed response task Set shifting and probabilistic reversal learning tasks Intertemporal choice task	=Motivation to work for food ↓ Working memory =Cognitive flexibility =Decision making	-	[153]
Chronic vapor exposure (10 mg/4 animals, PND 28–42 every other day, inh.)	Adolescent male Sprague-Dawley rats	Novel object recognition Conditioned avoidance response	=Discrimination index (↓ recognition memory when combined with alcohol adolescent exposure) No significant differences	-	[154]
Chronic vapor exposure (20 min of vapor exposure for 5 days)	Adolescent male Wistar rats	Fear conditioning	↑ Freezing time in Δ9-THC and Δ9-THC+ Ethanol CIE animals	↑ PrL signaling in response to the shock stimuli	[155]
Chronic (2.5 mg/kg PND 35–37, 5 mg/kg PND 38–41; 10 mg/kg PND 42–45, twice daily, i.p.)	Adolescent female Sprague-Dawley rats	Novel object recognition (classic and spatial versions)	↓ Recognition and spatial memory (discrimination index)	↑ Iba1, TNFα, COX-2, iNOS, and CB2 in the PFC ↓ CB1r and IL10 in the PFC	[156]
		-	-	↓ CB1r binding in the PFC Impaired eCB-LTD in the mPFC ↓ Dendritic length in DCX+ cells in the DG of the HIPP ↓ DCX+ cells in the DG of the HIPP Impaired newborn neurons-dependent LTP in the DG of the HIPP	[157]
	Adolescent male Sprague-Dawley rats	NOR (classic and spatial versions)	↓ Recognition and spatial memory (discrimination index)	↑ SYP, PSD95, GluA1, GluA2, GluN2B in the HIPP ↑ GFAP, TNFα, iNOS ↓ IL10 in the HIPP ↑ COX2 in the PFC	[158]
		Social interaction test Novel object recognition Prepulse inhibition	↓ Social recognition memory ↓ Object recognition memory ↓ % of prepulse inhibition	↑ The activity of VTA DA and PFC pyramidal neurons ↓ GSK-3 (α and β isoforms) protein expression in the PFC ↓ Phosphorylated AKT protein expression in the PFC	[159]
	Adolescent male Long-Evans rats	Prepulse inhibition Paired-associates learning	Impaired sensorimotor gating (4 months after Δ^9^-THC exposure) Delayed acquisition of learning criteria No differences in task performance	-	[160]
Chronic, (0.3 mg/kg PND 35–37; 1 mg/kg PND 38–41; 3 mg/kg PND 42–45; i.p.)	Adolescent male Sprague-Dawley rats	Prepulse inhibition Attentional set-shifting test Novel object recognition (spatial version) Morris water maze	=% of prepulse inhibition No significant differences ↓ Discrimination index ↓ Spatial memory	↑ CB1r protein expression in the PFC ↓ BDNF protein expression in the PFC and HIPP ↑ TrkB protein expression in the HIPP Impaired dopaminergic activity in the HIPP, PFC, dorsal striatum and NAcc ↓ SOX2+ and DCX+ cells in the HIPP (impaired hippocampal neurogenesis)	[164]
Chronic Adolescence (2.5 mg/kg PND 30–32; 5 mg/kg PND 33–36; 10 mg/kg PND 37–41; i.p.) Late adolescence (2.5 mg/kg PND 45–47; 5 mg/kg PND 48–51; 10 mg/kg PND 52–56; i.p.)	Adolescent male Sprague-Dawley rats	Novel object recognition (spatial version) Novel object recognition (classical version) Water T-maze	↑ Discrimination index in late adolescent-treated rats ↓ Discrimination index in adolescent-treated rats ↑ Number of trials to reach learning criteria	Impairment of vSub-NAc LTP amplitude	[165]
Δ^9^-THC pure (2.5 mg/kg PND 35–37; 5 mg/kg PND 38–41; 10 mg/kg PND 42–45; i.p.) Δ^9^-THC-rich/CBD-poor (THC 2.5 mg/kg + CBD 0.83 mg/kg PND 35–37; THC 5 mg/kg + CBD 1.66 mg/kg PND 38–41; THC 10 mg/kg + CBD 3.32 mg/kg PND 42–45; i.p.) CBD-rich/THC-poor (THC 0.15 mg/kg + CBD 5 mg/kg PND 35–45, i.p.).	Adolescent female Sprague-Dawley rats	Novel object recognition	↓ Object recognition memory (discrimination index)	↓ CB1r protein expression (Δ^9^-THC pure) in the PFC ↓ GAD67 (Δ^9^-THC pure) and ↑ GAD67 (Δ^9^-THC-rich/CBD-poor, CBD-rich/THC-poor) protein expression in the PFC ↑ CD11b (Δ^9^-THC pure, Δ^9^-THC-rich/CBD-poor) in the PFC	[161]
Adolescent chronic treatment (2.5 mg/kg, PND 35–37; 5 mg/kg, PND 38–41; 10 mg/kg, PND 42–45; i.p.) Adult chronic treatment (2.5 mg/kg, PND 75–77; 5 mg/kg, PND 78–81; 10 mg/kg, PND 82–85; i.p.)	Adolescent and adult female Sprague-Dawley rats	Novel object recognition	↓ Recognition memory (discrimination index)	Impairment of histone modifications (mainly H3K9me3) and expression of plasticity genes (more widespread and intense after adolescence treatment) in the PFC	[162]
Chronic (2.5 mg/kg from pnd 28 to 34; 5 mg/kg from pnd 35 to 40; 10 mg/kg from pnd 41 to 45; i.p.)	Adolescent male and female Wistar rats	Novel object recognition Prepulse inhibition	↓ Recognition memory (discrimination index) in THC-treated female rats No significant effects	↓ Leptin plasma levels in THC-treated female rats ↓ Arc in the PFC of THC-treated male and female rats ↓ Prepro-orexin in the Hyp of THC-treated male rats	[166]
Chronic smoke exposure (5 cigarettes/day from PND 29 to 49, 5.6% of Δ^9^-THC, inh.) Chronic (2.5 mg/kg, PND 35–37; 5 mg/kg, PND 38–41; 10 mg/kg, PND 42–45; i.p.)	Adolescent male and female Long-Evans rats	Novel object recognition	No significant differences in the discrimination index	-	[163]

5HT2Ar: serotonin receptor 2A; Δ^9^-THC: Delta-9-tetrahydrocannabinol; A2Ar: adenosine receptor 2A; Arc: activity-regulated cytoskeletal-associated protein; BDNF: brain-derived neurotrophic factor; CA: cornus ammonis subregion of the hippocampus; CB1r: cannabinoid receptor 1; CB2r: cannabinoid receptor 2; CD11b: integrin alpha-M; COX-2: cyclooxygenase 2; D2r: dopamine receptor type 2; D3r: dopamine receptor type 3; DA: dopamine; DCX: doublecortin; DG: dental gyrus of the hippocampus; DH: dorsal hippocampus; DRT: delayed response task; DN-DISC1: dominant-negative disrupted-in-schizophrenia-1; eCBs: endocannabinoids; GABABr2: gamma-aminobutyric acid (GABA) B receptor 2; GAD67: glutamic acid decarboxylase 67; GAT-1: GABA transporter type 1; GluA1: glutamate ionotropic receptor AMPA type subunit 1; GluA2: glutamate ionotropic receptor AMPA type subunit 2; GFAP: glial fibrillary acidic protein; GluN2A: glutamate ionotropic receptor NMDA type subunit 2A; GluN2B: glutamate ionotropic receptor NMDA type subunit 2B; GSKα/β: glycogen synthase kinase 3 α/β; H3K9me3: histone 3 lysine 9 trimethylation; Hyp: hypothalamus; HIPP: hippocampus; Iba1: ionized calcium-binding adapter mole-cule 1; IL: infralimbic cortex; IL10: interleukin 10; iNOS: inducible nitric oxide synthase; LTD: long-term depression; LTP: long-term potentiation; mPFC: medial prefrontal cortex; -: not available; NAcc: nucleus accumbens; NCAM-180: neural cell adhesion molecule isoform 180; NF-kB: nuclear factor kappa B; NOR: novel object recognition; pAktSer473: Akt phosphorylation on Ser473; pERK1-2: phosphorylated extracellular-signal-regulated kinase 1-2; PFC: prefrontal cortex; pGSK3a: glycogen synthase kinase 3 alpha; PKC: protein kinase C; pmTOR: phosphorylated mammalian target of rapamycin; PND: postnatal day; pNR2B: phosphorylated NR2B subunit of the N-methyl-D-aspartate receptor (NMDA); polySia: polysialic acid; PR: progressive ratio; PrL: prelimbic cortex; PSD95: postsynaptic density protein 95; RTT: rotating turntable task; SOX2: sex determining region (SRY)-box transcription factor 2; St8sia2: ST8 alpha-N-acetyl-neuraminide alpha-2,8-sialyltransferase 2; SYP: synaptophysin; TNFα: tumor necrosis factor alpha; TrkB: tropomyosin receptor kinase B; vHipp: ventral hippocampus; VTA: ventral tegmental area.

### 3.3. Psychostimulant Use Disorders

#### 3.3.1. Clinical Studies

##### Cocaine Use Disorders

Cocaine blocks the monoamine transporter, inhibiting the reuptake of dopamine, serotonin, and norepinephrine, leading to increased dopaminergic transmission in the mesolimbic pathway, amygdala and frontostriatal circuitry [171,172]. Furthermore, in individuals with active drug use, volumetric reductions in basal ganglia or thalamus and microstructural alterations have been reported in white matter (left superior longitudinal fasciculus, right cingulum, right hippocampal cingulum, forceps minor, and uncinate fasciculus) and gray matter (frontal and parietal-temporal areas) [173]. However, long-term neurotoxicity in animals and humans indicates that its use does not lead to a dopaminergic decrease in the structures but to macrostructural changes, which could be reversible when discontinued [172]. Although longitudinal studies are not very numerous, Parvaz (2022) reported an increase in baseline gray matter in ventromedial, orbitofrontal, and inferior frontal gyrus areas that are significantly related to better performance in cognitive flexibility and decision-making tasks (WCST and Iowa Gambling Task (IGT)) in patients with intermediate abstinence [174].

However, using functional neuroimaging, even patients with prolonged abstinence show a differential activation pattern in the frontoparietal network during motivational processing compared to non-users. The greater activation detected in these areas was related to a more significant number of years of use [175], or during cognitive control and interference management processes [176]. Differences in the functional activation of the frontoparietal network seem to be related to difficulties in goal-directed cognitive control processes, which nowadays are considered inherent to addictions, not only to cocaine use. In cocaine use disorders (CoUD), it appears that it may predict pathological use in young users and relapse [177,178]. Nevertheless, longitudinal and cross-sectional evaluations suggest that some cognitive, behavioral and neurofunctional control functioning recovery may be detected after abstinence [179].

Despite contradictory data, acute cocaine use produces alterations, or at least different processing in attention, psychomotor speed, reward management, emotional regulation, and executive functions [180]. Multiple altered components have been described in connection with executive functions. These include inhibitory processes (inhibitory control and impulsivity), cognitive flexibility, and working memory, usually assessed with the color-word Stroop test (to evaluate attentional deficits), go/no-go tasks, the Wisconsin Card sorting test (WCST), or the Iowa gambling task (IGT). However, not all studies support the idea that these impairments are homogeneous or persistent over time despite abstinence or even that they are being assessed correctly [172,181,182]. In addition, a large number of factors influencing cognition in CoUD have to be taken into account, including the influence of premorbid IQ level, the pattern of substance use, the existence of poly-drug use or the presence of emotional disorders such as stress, depression, and even sleep disorders [180].

The literature reports scattered neuropsychological data during acute use that could be related to the dose of cocaine consumed [181]. For example, Fillmore (2005) found that people consuming 200 mg to 300 mg of cocaine show better inhibitory capacity during go/no-go tasks [183]. Therefore, they conclude that cocaine administration reduces the time required for response inhibition, improving success in inhibiting incorrect responses. Indeed, for intakes of doses of 40 mg/70 kg body weight, Garavan found better inhibitory control when acute use increased activation of medial and lateral areas of the CPF [184]. However, these areas are down-regulated in long-term chronic users, which may account for cognitive failures in these people [185].

Jiménez and colleagues (2019) wanted to classify people as CoUD or non-CoUD using matching learning, according to the neuropsychological profile, to explore a more comprehensive range of cognitive functions. They used two datasets of active users at the assessment time and healthy non-users. The assessment included tests of cognitive flexibility (Berg’s card sorting test, BCST), inhibition (Flanker task and go/no go task); working memory (letter-number sequencing and digit span backward); decision making (IGT), Planning (Tower of London), and theory of mind (reading the mind in the eyes test, RMET). They conclude that some neuropsychological variables allow a person to be classified as CoUD, including cognitive flexibility, decision-making, inhibition, and theory of mind [182].

Some deficits in specific cognitive processes have been reported in the literature. However, there are few studies involving pure cocaine users or follow-up studies to estimate the impact of cognitive impairment. With this in mind, a different performance profile in attentional function has been found in CoUD [186]. For example, lower attention span is one of the measures altered in users with a short abstinence time, at baseline and after 4 weeks [187], and in CoUD with up to 8 months of abstinence, irrespective of the age of onset of use [186]. However, Chao and colleagues found that in users aged over 50 years with 5 days of abstinence, who began using after age 20 years and who maintained frequent use for at least 15 years before the assessment, there are no significant attentional differences compared to a control group of healthy individuals. Although the neuropsychological assessment performed is similar to that of Lopes [187] or Almeida [186], the sample assessed by Chao is smaller [188]. Therefore, Chao’s findings suggest that frequent cocaine use does not lead to more significant cognitive impairment than expected for age [188].

The learning and memory process has also been assessed in patients with CoUD. In the work of Almeida and colleagues [187], they found that patients with CoUD have a lower performance in memory and learning before and after treatment; however, an improvement in verbal learning measures has been observed in patients after four weeks of treatment, although the improvement does not reach the level of healthy people. In addition, they found that some variables related to the use of cocaine showed a significant effect on performance in learning ability. In early abstinence, learning scores (in recognition and long-term recall) were correlated negatively with the duration of cocaine use. After one month of abstinence, worse performance in total learning was associated with the age of onset of cocaine use. Lopes and colleagues conducted a study comparing early-onset (<18 years of age), late-onset (>18 years of age) and healthy subjects, finding that early-onset users differ significantly from healthy controls in terms of working memory. In contrast, late-onset participants did not present such a difference [186]. These results support the hypothesis that early-onset consumption impacts working memory and learning impairment, and it becomes more significant with more years of consumption [188] and entails more significant difficulties in retrieving processes such as working memory [189]. These results support the hypothesis that substance use in neurodevelopmental stages has more significant cognitive consequences than substance use in adults.

Blanco-Presas and colleagues [190] used the Rivermead Memory Test (RBMT) to assess the influence of CoUD in alcohol drinkers, finding that those with CoUD and AUD achieved lower scores in all areas assessed by the test, including prospective memory, visual memory, recognition, and immediate and delayed recall compared to non-drinkers. The authors attribute these results to the effect of alcohol, as the groups of consumers differed only in visuospatial and perceptual performance and verbal fluency. In contrast, cocaine users performed similarly to healthy people [190].

The effect of cocaine on different memory components is one of the most consistent findings independent of withdrawal time but possibly linked to alcohol consumption. None of the reported studies excluded participants who drank alcohol from their samples.

As noted above, impairments in different components of executive function have been extensively assessed, reflecting the clinical and functional repercussions they have in the lives of people using other drugs, specifically cocaine. Using general measures of executive function such as the Frontal Assessment Battery, a decrease in performance was found in cocaine and alcohol users [190], independent of the age of onset of use [186]. Indeed, poorer performance in cognitive flexibility, decision-making, inhibition, and theory of mind allows people to be classified as CoUD [182].

One of the most relevant considerations for understanding substance use is inhibitory control, involving the ability to inhibit a response (cognitive or motor) when it is not appropriate. Specifically studying this component, Czermainski and colleagues conducted a systematic review to determine any alteration in the inhibitory control in crack and/or cocaine users. They concluded that 90% (final n = 36) of the reviewed studies reported alterations in this cognitive component with a reduced inhibitory control, assessed using the go/no go, stop task, or color-word Stroop test. However, they suggest that most studies include participants who consume other psychostimulants [191], reflecting the difficulty in finding studies with pure cocaine users. Further studies showed better inhibitory performance after acute cocaine administration [183,192].

Despite the results obtained through neuropsychological tests, functional neuroimaging techniques suggest a pattern of neurofunctional compensation, with a more significant effort to achieve effective inhibitory control results. In this sense, the long-term activation of brain areas related to inhibitory control (right frontoparietal network and amygdala-striatal network) seems to be reduced in patients with CoUD. However, abstinence after treatment has been associated with greater activation in these networks during interference control tasks such as the Stroop [3].

Therefore, more significant inhibitory capacity and superior cognitive flexibility are significantly related to the effective completion of treatment programs and increase therapeutic adherence [193].

The decision-making process has been one of the most evaluated constructs in addictions [194,195]. It involves selecting the most appropriate response according to the circumstances surrounding the patient. The selection of the most suitable choice evaluates the value rewards correctly, considering the negative consequences of the behavior, the immediate reinforcement, and the inhibition of automatic responses. One of the tests commonly used by researchers in the IGT aims to assess reward-delaying ability and decision-making style. In this test, different results are related to active and long-term consumption, evaluated under abstinence. Cocaine users tend to present a response profile that differs from healthy people. For instance, worse performance is found in the IGT after one month of abstinence in pure users, which is more significant, as the history of previous use was worse (higher dose, longer duration of use) [196]. However, the higher the total IGT score during acute use, the higher the probability of CoUD [182]. The authors explain this result by mentioning that although the test is performed better, the responses carried out by the CoUD group are riskier. Thus, the greater the number of risky responses, the greater the gain [182,197].

Despite these results, differential decision-making performance is not specific to cocaine users. No data are available to support the idea that different drugs of abuse affect decision-making but rather that they are a common feature of all addictions [194], suggesting that this cognitive factor could appear before the development of addictive behavior.

#### 3.3.2. Preclinical Studies

Chronic cocaine consumption is associated with abnormal brain morphology [198], which causes executive function, memory, language, and psychomotor speed problems. Moreover, it carries a high risk of suffering from vascular disease [199]. Several animal studies demonstrated cognitive impairment caused by chronic use of cocaine. However, few animal studies show significant memory consolidation changes following acute cocaine administration. It is essential to mention that this damage is long-lasting, even long after the discontinuation of drug use. This section highlights animal models’ main cognitive alterations after acute and chronic consumption (Table 3).

Even long after the discontinuation of cocaine use, there are deficits in frontocortical function and cognition. In one study, rats were allowed access to cocaine for either 1 h/day (short access; ShA) or 6 h/day (long access; LgA) for three weeks. During the withdrawal period, early after the discontinuation of drug use, LgA animals were impaired on the sustained attention task with a significant decrease in the vigilance index. They continued to show patterns of performance deficits of disruption of cognitive flexibility even after 30 days of discontinued drug use. These cognitive alterations were associated with significant decreases in dopamine 2 receptor (*D2*) gene expression in the medial and orbital PFC and D2 protein expression in the medial PFC in LgA animals. However, no differences were found in ShA animals [200]. This result is consistent with previous reports by Calu and colleagues. A cocaine self-administration procedure in rats for 14 consecutive days found severe alterations in reversal learning even a month after withdrawal and a time-dependent increase in cocaine-seeking induced by a cue [201].

Several animal studies have investigated alterations in brain connectivity that may explain this cognitive impairment. These studies suggest chronic cocaine use is associated with altered brain connectivity between different structures, including cortical-striatal regions and the default mode network. Some molecular mechanisms could be involved in learning impairment. CB1 expression in cortical glutamatergic neurons may control associative learning processes in cocaine self-administration in mutant mice [202]. Another preclinical study with Long Evans rats measured the alterations in the mesocorticolimbic circuit by functional magnetic resonance imaging (fMRI) after cocaine self-administration paradigms in the abstinence period. Authors reported that resting-state functional connectivity (rsFC), as a measure of intrinsic neurobiological interactions, decreased in cocaine self-administration compared with the control group, between the prelimbic cortex and entopeduncular nucleus and between the NAcc and dorsomedial PFC [203]. Following a long period of abstinence from self-cocaine administration, three female rhesus monkeys were acutely administered cocaine and then evaluated by fMRI for global functional connectivity. Acute cocaine administration during abstinence decreased global functional connectivity and selectively impaired prefrontal circuits controlling behavior [204]. However, in one more recent study, Long Evans rats self-administered cocaine intravenously for 6 h sessions daily over 14 consecutive days. After one day of abstinence from cocaine, there was a high clustering coefficient in the amygdala, hypothalamus, striatum, HIPP, and thalamus.

In contrast, decreases were observed at 14 days of abstinence, measured by fMRI [205]. Thus, cocaine can significantly change the functional connectivity of brain regions involved in cognition and emotion during withdrawal, particularly early stages. These brain functional disturbances could be related to an enhanced cognitive ability focused on the acquisition and use of the drug to avoid the negative consequences of withdrawal.

Other animal studies demonstrated that cognitive alterations could be produced when the animals have free access to the drug (self-administration paradigms) and when the administration is controlled. For instance, C57BL/6 mice receiving 30 mg/kg/day injections of either cocaine or saline for 14 days, followed by 2 weeks of the withdrawal period, resulted in impaired learning and working memory [206]. In another study, mice were administered cocaine 20 mg/kg/day, i.p., for 12 consecutive days and underwent behavioral assessment in drug-free conditions and unfamiliar environments after 24 days of abstinence. Cocaine withdrawal mice showed cognitive deficits in spontaneous alternation behavior measured with the Y-maze test and in-place recognition memory. They were most active in the forced swimming test when escaping from the water, although they showed normal immobility behavior [207]. Therefore, in both experiments, animals showed impairment in working memory.

The performance of marmoset monkeys was evaluated in the recognition memory task after acute and repeated exposure to cocaine (5 mg/kg, i.p) for seven consecutive days. Acute administration of cocaine improved the marmoset’s recognition memory, whereas it had a detrimental effect after repeated exposure [208]. Thus, depending on the cocaine administration schedule, opposite effects on spatial recognition memory can occur, with prolonged exposure associated with cognitive impairment. However, additional studies are needed to elucidate the underlying neurobiological mechanisms.

Crack cocaine is a highly toxic drug with great potential to induce dependence. Its effects were evaluated on rats’ spatial working memory and brain oxidative stress parameters. Adult male Wistar rats, trained in an eight-arm radial maze (8-RM), underwent five sessions of crack inhalation once a day and were evaluated in 1 h delay tasks 24 h after the last inhalation. Animals from the crack cocaine group showed more errors in the 1 h post-delay performance in the 8-RM when compared to the control. Therefore, the repeated inhalation of crack cocaine impaired long-term spatial working memory [209]. A similar study used the metabolite from the pyrolysis of crack cocaine, the anhydroecgonine methyl ester (AEME), to see its effects on spatial working memory. Rats who received acute intracerebroventricular AEME were tested 1 h after the 8-RM task. More errors were seen after 32 µg and 100 µg of AEME than in control animals [210]. These cognitive disturbances could be explained, at least in part, by the higher neurotoxicity potential of AEME compared with cocaine itself. Indeed, an enhanced oxidative state was found in the striatum of AEME-treated animals [210].

**Table 3 biomedicines-11-01796-t003:** Summary of the findings related to animal models’ cocaine-induced cognitive and biological changes.

COCAINE					
Drug and Pattern of Administration	Strain and Stage of Administration	Test	Behavioral Alterations	Biological Alterations	References
Cocaine self-administration 1 h/day (short access; ShA) 6 h/day (long access; LgA)	Wistar rats Adulthood	Sustained attention test	Impaired on the sustained attention task of the LgA animals. ↓ Vigilance index	↓ *D2* gene expression in the medial and orbital prefrontal cortex and D2 protein expression in the medial prefrontal cortex in LgA animals	[200]
Cocaine self-administration 14 days	Male Long-Evans rats Adulthood	Reversal learning	Alterations in reversal learning even one month after withdrawal	-	[201]
Cocaine self-administration	Glu-CB1r vs. GABA-CB1r C mice	Associative process	T-Maze Learning and Reversal	CB1r expression in cortical glutamatergic neurons controlled the associative learning processes	[202]
Cocaine self-administration	Long-Evans rats	fMRI	-	↓ Neurobiological interaction between prelimbic cortex and entopeduncular nucleus and NAcc and DMPFC	[203]
Acute cocaine administration in the abstinence period after cocaine self-administration	Female rhesus monkeys	fMRI	-	↓ Global functional connectivity in the prefrontal circuitry	[204]
Cocaine self-administration 14 days	Long Evans rats	fMRI	-	After 1 day of abstinence high clustering coefficient in the AMY, hypothalamus, striatum, HIPP, and thalamus After 14 days of abstinence ↓ clustering coefficient	[205]
30 mg/kg/day 14 days After 2 weeks during the withdrawal period	C57BL/6	Reversal learning task Three-choice serial reaction time taskDelayed matching-to-position task	Impaired learning and working memory	-	[206]
20 mg/kg/day, i.p. 12 days	C57BL/6 J	Y Maze Open field exploration Spontaneous behavior Forced swimming test	Cognitive deficits in spontaneous alternation behavior and place recognition memory	-	[207]
Acute and chronic 5 mg/kg, i.p 7 days	Marmoset monkeys	Spontaneous object location	Acute administration improved the marmoset’s recognition memory Detrimental effects after the repeated exposure	-	[208]
5 sessions of crack inhalation once a day	Wistar rats	Eight-arm radial maze	Impaired long-term spatial working memory	↑ AOPP and SOD activity	[209]
Acute i.c.v AEME	Wistar rats	Eight-arm radial maze	32 µg and 100 µg impaired working memory	↑ 100 μg i.c.v glutathione peroxidase	[210]

5HT2Ar. AEME: anhydroecgonine methyl ester, AOPP: oxidation protein products, CB1r: cannabinoid receptor 1, DAD1−/−: dopamine D 1 mutant rat, D2: Dopamine 2 receptor, DMPFC: the dorsomedial prefrontal cortex, HIPP: hippocampus, i.c.v: intracerebroventricular, NAcc: nucleus accumbens, SOD: superoxide dismutase fMRI: magnetic resonance imaging.

### 3.4. Opioid Use Disorders

#### 3.4.1. Clinical Studies

Opioid use is broadly associated with cognitive deficits during chronic use and after abstinence [211]. Recent reviews show cognitive alterations in learning and memory processes, attention and executive functions [211,212], with neurofunctional alterations similar to other substances. These alterations are consistent with dysfunctions in the frontoparietal networks or structural alterations in the frontal lobe [211].

Abuse and chronic use of heroin are associated with poor cognitive performance in attention and executive functions such as working memory, inhibition, and decision-making [212]. The use of heroin alone impairs learning, memory, and attentional recruitment by drug-related cues, leading to further craving and drug-seeking behavior [212]. Working memory errors are also present in heroin-dependent individuals under opioid maintenance treatment, though this effect was studied as possibly related to the medication [213].

Recent studies evaluating cognitive aspects of opioid use disorders involve psychopharmacological maintenance treatments [214,215,216,217]. Patients with opioid use disorder and on substitution treatment (methadone or buprenorphine) showed more errors and less strategy use in a reward-based learning task, reflecting possible deficits in learning generalization [217]. Although buprenorphine-based treatment appears to produce less cognitive impact [211], this result is not consistently replicated [218].

A recent meta-analysis assessed the impact of opioid use on cognitive performance by considering studies that included opioid use generically, without differentiating between substances. Their results indicate that, despite the dispersion and methodological difficulties, longer abstinence times lead to better cognitive outcomes but are not influenced by years of opioid use [219]. The effect of opioids was largest on psychomotor activity and visual memory, moderate for attention and memory, and small on executive function. However, these results did not include a differential analysis of factors such as duration of use or time of abstinence at the assessment time.

Cognitive impairment seems to be related to treatment outcomes, since lower inhibition function was a predictor of decreased length of stay in treatment, and higher attentional interference was associated with higher rates of dropouts [220]. Furthermore, mild cognitive impairment occurred in a considerable proportion (66% to 68% of the sample), as assessed by the NIH Toolbox for the Neurological and Behavioral Function [215]. This tool evaluates memory, processing speed, language, attention, and executive function, and it allows the creation of scores for detecting mild cognitive impairment (MCI). In these patients, MCI scores were associated with sociodemographic variables (over 50 years old), low educational attainment (less than high school), ethnicity (belonging to a minority) and clinical variables (presenting a head injury, poly-drug use over the last two years). In addition, MCI scores correlated with self-reported cognition items that closely reflect essential functions for everyday life (e.g., forgetting the name of common things) [215]. Participating in an adjunct cognitive rehabilitation program (together with methadone treatment and psychosocial interventions) improved working memory deficits in opioid use disorder and were associated with a lower number of positive urine tests for drug use [220].

Decision-making functions remain affected even one year after heroin abstinence. It was not feasible to adhere to strict inclusion criteria in the reviewed studies, and these were often poorly reported. For instance, a cognitive evaluation study carried out in 21 opiate-dependent individuals attending a clinical health program described a rate of heroin use of 23.8% in the previous 30 days (before the assessments), together with reports of other substances use (tobacco, cannabis, and/or alcohol) [217]. This may be an essential issue to consider when evaluating the cognitive effects of opioid and specifically heroin use, together with clinical aspects related to the addiction process, such as frequency of use, duration of use, age of onset and other clinical characteristics (psychiatric comorbidities, other drug use, etc.) and medication pharmacokinetics related to the administration route (intravenous or smoked opiates). These factors may affect bioavailability and direct and indirect effects on the nervous central system functioning. However, few studies have tried to evaluate and control for these aspects. For instance, heroin users that meet the criteria for clinical dependence and are abstinent less than 24 h show visual attention and cognitive flexibility alterations, slowing down Trail Making Test-B performance compared to healthy controls [221]. Clinical aspects of heroin use, such as the total duration of drug use, were related to working memory deficits [213]. Other studies report opiate use and a history of using other substances (alcohol, cannabis, stimulants) in a detailed manner. However, opiate use and its lasting effects can hardly be isolated [222]. Murray and colleagues studied the impact of cigarette smoking and added opiate dependence in individuals taking buprenorphine, an opioid substitution treatment. They found that opiate-dependent individuals who used tobacco had worse executive functioning and working memory than individuals with tobacco use alone (without other substances). The possible role of tobacco consumption and other substance use reported by patients in the past and recent periods before the evaluation is impossible to rule out when studying impairments in working memory, processing speed, and visuospatial skills [222].

Emotional regulation deficits can also significantly impact clinical aspects related to addiction. Namely, patients with opioid use disorder under methadone treatment manifested a predictive relationship between non-acceptance of emotions and coping motives for drug use [216]. An intact cognitive recognition and re-appraisal of one’s thoughts and emotions may be crucial for intervening in the person’s train of thought to avert possible consumption behavior or relapse. Perception and attention to one’s internal feelings and sensations are, in this way, relevant to the psychological processes of the addiction course.

#### 3.4.2. Preclinical Studies

Morphine is the most commonly studied opioid, describing the cognitive impairment associated with opioid abuse and dependence. Both acute and chronic administration of morphine induces significant impairments in different types of memory in animals, mainly rodents (Table 4).

A single dose of morphine decreased the acquisition of spatial memory in the Morris water maze test and memory consolidation in the step-through passive avoidance test, whether administered before [223,224,225] or after the training session [226]. The microinjection of muscimol, a GABA-A receptor agonist, in the CA1 field of the HIPP, increased the memory impairment of morphine, suggesting a synergistic effect. The CB1 cannabinoid receptor (CB1r) has been proposed to interact with GABAergic and opioidergic systems. Indeed, the microinjection of a CB1r-agonist or antagonist in the basolateral amygdala (BLA) increased and reversed the impairment of memory induced by muscimol and morphine [226]. The same authors previously demonstrated that the CB1r-agonist arachydonilcyclopropylamide (ACPA), administered in the BLA, potentiated morphine-induced memory consolidation impairment. This effect was blocked by administering a GABA-A receptor antagonist or NMDA receptor antagonist in the ventral tegmental area (VTA). These results suggest that the endocannabinoid, GABAergic- and glutamatergic systems play a role in modulating the cognitive acute impairment effects of morphine [227]. Additional studies that evaluate the detrimental cognitive effects of repeated morphine administration employing different paradigms are needed.

In rodents, chronic morphine induces spatial memory impairments and long-term hippocampal potentiation (LTP) [228,229]. Cognitive dysfunctions have also been observed in mice treated chronically with morphine in the social and object recognition tests, depending on the strain [230,231,232]. Similarly, repeated administration of morphine produced an impairment to attentional responses and processing speed with increased impulsivity in the five-choice serial reaction time task (5-CSRTT) [231,232,233].

These cognitive impairments were associated with a reduction of gene and protein expressions of postsynaptic density protein (PSD95) and cAMP response element-binding protein (CREB) phosphorylation in the PFC [231,232,233], two synaptic markers involved in synaptic changes [234,235] and the formation of LTP [236]. The regulation of PSD95 by CREB has been associated with synaptic transmission changes and rewarding behaviors induced by morphine conditioning [237]. Additionally, the extracellular accumulation of adenosine supports the involvement of the endogenous adenosine system in the effects of morphine in learning and memory [238].

Moreover, chronic morphine impaired memory in step-through passive avoidance in Wistar rats [239], increasing the expression of the peroxisome proliferator-activated receptor γ coactivator-1α(PGC-1α)—a master regulator of the mitochondrial biogenesis which activates antioxidant pathways [240,241,242]—and decreasing the expression of cocaine-amphetamine regulated transcript (CART)—closely related with the opioid mesolimbic dopamine system [243]. The impairment of morphine in this test depends on the time of administration [244]. An additional study in rhesus monkeys revealed the interaction between critical elements of the dopaminergic system and morphine in spatial working memory impairment [245].

Cumulative evidence supports the reduction of neurogenesis and BDNF, mainly in the HIPP, as one of the mechanisms by which morphine causes attentional and memory deficits. Ghodrati-Jaldbakhan and colleagues conducted a study on female rats, demonstrating how chronic morphine administration induced a detriment in object location memory and reduced hippocampal BDNF [232]. Similarly, a reduction of neurogenesis and BDNF levels was identified in isolated adult male rats treated with morphine, suggesting that isolation increased the detrimental effects of morphine on spatial memory [246]. More recently, research found that subchronic exposure to morphine impaired effort- and delay-based forms of cost-benefit decision-making, inducing significant changes in BDNF, p-CREB/CREB and p-GSK3β**/**GSK3β in the amygdala [247].

Acute and chronic morphine administration modified NMDA receptor subunits’ expression [248,249], crucially involved in drug-induced associative memories [250,251]. Morphine withdrawal was also associated with cognitive deficits. In rodents, spontaneous and induced-morphine withdrawal (with naloxone) induced short-term memory alterations in the NOR, increasing brain corticosterone concentrations [252].

In brief, acute and chronic administration of morphine impairs different types of memory, including spatial and working memories, in which CB1r, D1r, GABA, NMDA, BDNF, adenosine and CREB appear to play a relevant role.

**Table 4 biomedicines-11-01796-t004:** Summary of the findings on morphine-induced cognitive and biological changes in animal models.

MORPHINE					
Pattern, Doses and Route of Administration	Strain and Stage of Administration	Test	Behavioral Alterations	Biological Alterations	References
1, 3 and 10 mg/kg; i.p.; pre and post-training, acute	Adult male Swiss albino mice	Step-through passive avoidance test	No alteration in memory retention Impairment of memory retrieval	-	[223]
1 and 2.5 mg/kg; i.p.; post-training, acute	Adult male DBA/2 mice		Impairment of memory retrieval	-	[224]
3 and 5 mg/kg; i.p.; post-training, acute	Adult male Wistar rats		Impairment of memory retrieval	-	[226]
3 and 6 mg/kg; i.p.; post-training, acute	Adult male Wistar rats		Impairment of memory retrieval	-	[227]
2.5, 5 and 7.5 mg/kg; i.p., acute	Adult male Wistar rats	Morris water maze	↓ Spatial memory acquisition	-	[225]
0.01, 0.1 and 0.2 mg/kg; i.p., acute	Adult male Rhesus monkey		Impairment of spatial working memory		[245]
2.5, 5 and 7.5 mg/kg i.p.; 24 h/3 days; pre-training (morphine sensitization), subchronic	Adult male Wistar rats	Morris water maze	Reverses the impairment of morphine	-	[225]
Increasing doses (from 30 to 90 kg/kg s.c.; 12 h/3 days, subchronic	Male NMRI mice	Novel object recognition	Short-term memory alterations	↑ Brain corticosterone concentrations in the whole brain	[252]
Increasing doses from 5–50 mg/kg s.c.; 12 h/6 days, subchronic	Adult male Wistar rats	A cost-benefit decision-making T-maze	Impaired effort and delay-based form of cost-benefit decision making	BDNF, p-CREB/CREB and p-GSK3β/GSK3β in the AMY	[247]
10 mg/kg; s.c.;12 h/10 days, chronic	Adult male Sprague-Dawley rats	-	-	↓ LTP in CA1	[228,229]
3 mg/kg; i.p.; 12 h/14 days	Adult male C57BL/6 and BALB/cJ mice	Novel object recognition Social recognition test	↓ Discrimination ratio for both strains in novel object recognition ↓ Discrimination ratio of C57BL/6 in the social recognition test	-	[230]
10 mg/kg; i.p.; 24 h/14–17 days	Adult male C57BL/6J mice	5-choice serial reaction time task	↓attentional responses ↓processing speed ↑ impulsivity	↓ PSD95 and ↓ CREB in PFC and HIPP	[231,233]
Increasing doses from 2 to 30 mg/kg i.p.; 21 days	Adult male Wistar rats	Step-through passive avoidance test	Impairment of memory	↑ PGC-1α and ↓ CART in the HIPP	[239]
10 mg/kg; i.p.; 12 h/10 days	Female Wistar rats	T-maze Step-through passive avoidance test NOR	Impairment in all tests	↓ BDNF in HIPP	[232,253]
0.75 mg/rat; i.p.; 14 days	Adult male Sprague-Dawley rats	Morris water maze	Impair spatial memory	↓ neurogenesis ↓ BDNF in HIPP	[246]

BDNF: brain-derived neurotrophic factor; CART: cocaine-amphetamine regulated transcript; CREB: cAMP response element-binding protein; CPu: caudate-putamen; D1r: dopamine receptor 1; D3r: dopamine receptor 3; HIPP: hippocampus; i.p.: intraperitoneal administration; LTP: long-term potentiation; MMP: mitochondrial membrane potential; NAcc: nucleus accumbens; NMRI: Albino Naval Medical Research Institute mice; ROS: reactive oxygen species; PFC: prefrontal cortex; PGC-1α: peroxisome proliferator-activated receptor γ coactivator-1α; PSD95: postsynaptic density protein; s.c.; subcutaneous administration.

### 3.5. Molecular Alterations Associated with Cognitive Decline in Addicted Patients

This section describes some central and peripheral molecular changes potentially involved in cognitive decline after different types of drug consumption. Despite the diversity of drugs of abuse acting on distinct molecular targets, all activate the brain reward circuits, inducing adaptative changes in several brain regions. These changes correlate with psychological and physical drug effects and cognitive alterations during consumption and withdrawal (Figure 2).

#### 3.5.1. Dopaminergic System

Striatal dopamine activity contributes to cognitive flexibility, an essential cognitive strategy to inhibit impulsivity and compulsive drug-seeking behaviors. Changes in different components of the dopaminergic system were found mainly in patients with stimulant use disorders, such as methamphetamine (METH) or cocaine, and alcohol and opioid addicts [254]. Dopaminergic D2/D3 receptor availability in the striatum was associated with executive function in healthy individuals, and in patients with drug addiction this could contribute to impaired executive functioning [255]. A PET study with METH users revealed significantly lower striatal D2/D3 receptors than controls. However, these changes appear to be associated with healthy controls’ executive functions but not METH users’ [255].

Another study reported that cocaine users presented disrupted brain dopaminergic neurotransmission. Cocaine abusers and healthy controls were exposed to repetitive cycles of the color-word Stroop task, evaluating mental fatigue by increased errors and decreased post-error slowing. The fMRI evaluation showed disrupted D2 receptor availability in caudate-putamen and reduced dopaminergic activity in the midbrain of addicted patients. The administration of methylphenidate, a dopamine transporter inhibitor, reversed these brain responses, restoring the dopaminergic signaling in cocaine users [256]. Peripheral mRNA levels of dopamine receptors were also evaluated in patients with stimulant use disorders and cognitive impairment. Patients with these disorders performed worse on the cognitive tests assessing working memory.

METH consumption causes a significant release of dopamine, associated with neurotoxic and executive dysfunction in METH users. The catechol-o-methyltransferase gene (*COMT*) degrades dopamine and other monoamine neurotransmitters. The Val allele of a single nucleotide polymorphism (SNP) of the *COMT* gene (Val158Met) confers more significant dopamine degradation in the PFC than the Met allele. Cherner and colleagues evaluated the role of this SNP in executive functioning in METH users. The authors found differences in executive function only among Met/Met carriers but not in Val carriers. Therefore, the increased significance of DA inactivation of the Val allele may protect against METH-related executive dysfunction, suggesting genetic differences in vulnerability to METH [257].

The effects of the same SNP with another one affecting D2 receptors were evaluated on reversal learning after the acute administration of cocaine or cannabis. Authors showed that the results of cocaine but not cannabis on reversal learning depend on individual genetic differences in the dopamine D2 receptor gene [258]. Despite that, the functional SNP Val158Met appears to influence the immediate cognitive and psychotic effects of cannabis. Indeed, Δ^9^-THC administration to patients with previous cannabis consumption and psychotic symptoms impaired working memory performance in COMT Val/Val but not in Met carriers, suggesting the involvement of this SNP in cognitive function [259].

Dopaminergic D2 receptor functioning has been associated with AUD and related phenotypes, including executive functioning. Hagerty and colleagues [260] evaluated the relationship between a continuous measure of alcohol-related problems, epigenetic markers (methylation) within the D2 gene, and functional connectivity within the executive control network (ECN) among polydrug users. They found that the methylation of the *D2* gene was negatively associated with right and left ECN and significantly associated with the severity of alcohol-related problems [260]. The association between the SNP of the dopaminergic *D2/ANKK1* gene (rs1800497), the serotonin-related polymorphism in the *5HT2A* gene (rs6313), and impulsivity, cognition, and alcohol misuse in young adults was studied. However, no correlations were found between these parameters [261]. Alcohol-dependent patients showed reduced striatal dopamine densities and responded differentially to dopaminergic drugs. Indeed, in an alcohol-dependent patient population, apomorphine (dopamine agonist) administration improved the AX-continuous performance test compared to controls. This may imply multiple cognitive domains (including inhibition, attention, and working memory), suggesting this system has a role in alcohol-induced cognitive impairment [262].

Some studies have also analyzed the contribution of this system to the cognitive decline induced by the repetitive consumption of opioids. One described a linear correlation between the striatal dopamine transporter availability and cognitive flexibility in opioid-dependent patients [221].

Thus, considering the available information, the dopaminergic system seems to be intensely involved in the cognitive performance of drug abusers. However, further studies are required to analyze and improve our understanding of its role in this complex clinical situation.

#### 3.5.2. Brain-Derived Neurotrophic Factor

BDNF is one of the most common neurotrophic factors, intensely involved in learning and memory [263]. In the adult brain, BDNF preserves high expression levels and regulates excitatory and inhibitory synaptic transmission. High levels of BDNF have been detected in the hippocampus, amygdala, cerebellum, and cerebral cortex in humans, with the highest levels in hippocampal neurons [264]. Nerve growth factor (NGF) is implicated in the maintenance of basal forebrain cholinergic neurons in adulthood. The NGF metabolic pathway is impaired in Alzheimer’s disease and other amyloid pathologies [265]. However, considering that BDNF expression is highly altered in structures essential for memory processes, such as HIPP and para-hippocampal areas, our study focused on BDNF changes in substance use disorders associated with cognitive impairment [264].

Circulating levels of BDNF were analyzed in several studies, searching for a potential relationship with cognitive functioning. In a study by Requena-Ocaña and colleagues, BDNF and neurotrophin-3 concentrations were related to the cognitive reserve and the individual’s ability to manage a brain disease through compensatory mechanisms of cognitive stimulation. Lower educational levels were associated with the early onset of alcohol consumption, development of alcohol dependence, and impaired frontal cognitive functioning [266]. On the other hand, circulating BDNF levels also correlated with a frontal assessment battery in alcohol-abstinent participant [267], as well as with lysophosphatidic acid (LPA) species concentrations in abstinent alcoholic patients. LPAs are bioactive lipids involved in several neurological processes, including neurodevelopment and addiction. These patients had decreased LPA levels associated with dysexecutive syndrome and memory impairment. In addition, BDNF concentrations are related to total LPA concentrations and cognitive dysfunction [268].

BDNF has also been associated with METH dependence. Some studies have analyzed its involvement in METH-induced cognitive decline. High concentrations of BDNF were found in METH-dependent patients, associated with a delayed memory index score. The repeatable battery for the Assessment of Neuropsychological Status (RBANS) total score, immediate memory, or attention index was positively correlated with the Val/Val alleles of the *BDNF* gene. However, a negative association was reported between RBANS visuospatial/constructional or language index and Met/Val patients [269]. In the abstinence phase, a decrease in mBDNF, proBDNF, and MMP-9 concentrations was found in cognition-impaired METH-dependent patients in early withdrawal. Thus, these parameters could be potential biomarkers of cognitive decline in METH abusers [270]. However, this decrease in BDNF concentrations was not found in crack/cocaine-addicted women in the abstinence phase. In this case, the concentrations of BDNF were higher and associated with delayed verbal recall [271]. These discrepancies should be considered with caution due to the timing of abstinence at the time of the evaluation, which may be an additional variable to consider in future studies.

Other studies evaluated *BDNF* or *AKT1* gene SNP concentrations in heroin- or cannabis-dependent patients and their association with attention and language index [272,273], finding no association between these parameters, which suggests that changes in *BDNF* and related genes could be more specific to alcohol or stimulant use.

#### 3.5.3. Opioid System

The endogenous opioid system plays a critical role in addictive disorders. The dynorphin-kappa-opioid receptor (DYN/KOR) system may undergo adaptive brain changes, which, along with neuronal loss, may contribute to alcohol-associated cognitive decline. Alterations in different system components were evaluated in brain samples of alcoholic patients. Prodynorphin (*PDYN*) gene expression and DYN levels in the dorsolateral prefrontal cortex (DLPFC), *KOR* gene expression in the orbitofrontal cortex (OFC), and DYNs in the HIPP were up-regulated in alcoholics. The activation of KORs could underlie, at least in part, the neurocognitive dysfunctions relevant to addiction and disrupted inhibitory control [274]. DYNs mediate alcohol-induced learning and memory impairment, while KOR antagonists block excessive, compulsive-like drug and alcohol self-administration in animal models.

*PDYN* down-regulation was found in people with a high-risk C allele of *PDYN* promoter SNP rs1997794 associated with alcoholism. Thus, changes in *PDYN* expression may be attributed to transcriptional adaptations in the alcoholic brain.

#### 3.5.4. Serotonergic System

The serotonergic system is altered by cocaine or methylenedioxymethamphetamine (MDMA or ecstasy) consumption and is also involved in cognitive performance. Cognitive deficits were found in MDMA users associated with reduced brain serotonergic (5HTT) but not dopaminergic (DAT) transporter densities. Indeed, memory performance was associated with 5HTT binding in the DLPFC, OFC, and parietal cortex, brain regions involved in memory function. However, MDMA consumption significantly decreases the strength of this relationship, suggesting its role in cognitive impairment [275]. Both ecstasy consumers and ex-consumers show significantly impaired verbal recall, with a reduced 5HTT availability in the mesencephalon and caudate nucleus. These data suggest the involvement of the serotonergic system in memory impairment in current consumers and MDMA-abstinent patients [276]. Roiser and colleagues investigated whether the s allele of the *5HTT* gene polymorphic region (5HTTLPR), associated with reduced serotonergic transmission relative to the l allele, relates to the increased vulnerability to MDMA on cognitive function. In controls, ss and ls genotypes were associated with attention and decision-making. However, in MDMA consumers, this relationship was attenuated. These results are consistent with the idea that cognitive impairment in ecstasy users could depend on genetic variation at the 5HTTLPR [277].

The same polymorphism was assessed in chronic cocaine users, evaluating the impact of the genetic variations affecting 5HTT activity and the peripheral *5HTT* gene expression on working memory performance. The long/long (5HTTLPR) and C/C (TPH2 rs1386497 of tryptophan hydroxylase) genotypes were risk alleles for working memory impairment. In healthy controls, these polymorphisms were associated with improved working memory performance. These changes suggest that the serotonergic system is essential in developing cognitive deficits in chronic cocaine users [278]. Cannabinoid CB1 and serotonergic 2A receptors form heteromers that may be involved in the cognitive deficits produced by Δ^9^-THC administration. In cannabis consumers, these heteromers significantly increased in the olfactory neuroepithelial cells, showing a positive association with the amount of cannabis consumption and a negative one with the age of onset of cannabis use. The authors concluded that cannabis consumption regulates the formation of these heteromers and may play a critical role in cognitive processing [279].

#### 3.5.5. Immune System

A few studies have evaluated the immune system’s implication in cognitive impairment induced by drugs of abuse, revealing the potential significance of both pro- and anti-inflammatory cytokines in several cognitive domains. However, the relationship between inflammation and cocaine use is more complex, and further studies are required to clarify its role. In crack/cocaine-dependent women, higher plasmatic interleukin 6 (IL-6) concentrations were associated with lower executive functioning. IL-6 levels were analyzed in alcohol-dependent patients at the onset of the detoxification period and 21 days later. A higher degree of neuroinflammation was associated with more pronounced cognitive impairment [280]. However, this direct correlation was not observed in opioid-dependent patients with methadone treatment. In this study, changes in TNF-alpha and IL-6 levels were negatively correlated with the changes observed in the verbal index [281].

**Figure 2 biomedicines-11-01796-f002:**
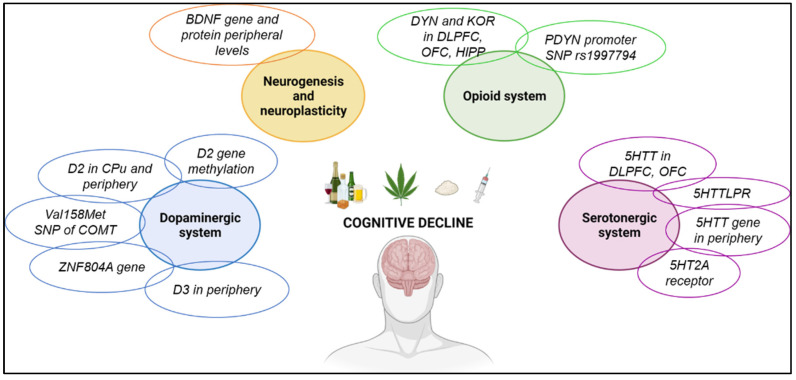
A summary of the molecular alterations associated with cognitive decline in patients with substance use disorders. 5HTT: serotonin transporter; 5HTTLPR: *5HTT* gene polymorphic region; 5HT2A: serotonergic receptor 2A; BDNF: brain-derived neurotrophic factor; COMT: catechol-o-methyltransferase gene; CPu: caudate putamen; D2: dopaminergic receptor 2; D3: dopaminergic receptor 3; DLPFC: dorsolateral prefrontal cortex; DYN: dynorphin; HIPP: hippocampus; KOR: kappa opioid receptor; OFC: orbitofrontal cortex; PDYN: prodynorphin; SNP: single nucleotide polymorphism.

## 4. Discussion

A large body of evidence exists about different implications of the neurotransmitter system in the cognitive decline induced by drug consumption. Differences mainly relate to the type of drug abuse and the phase of addiction in which the evaluation is performed. However, the results are inconsistent in some cases, and more studies are required to further explore the molecular changes underlying drug-induced cognitive impairment.

This review summarizes the main findings on cognitive alterations in drug-dependent patients and the associated molecular alterations from a translational approach. The connection between cognitive impairment and drug consumption is well known. These alterations may occur after acute drug exposure. However, chronic exposure usually induces more pronounced disturbances with greater brain effects. In some cases, abstinence after chronic drug use is also associated with learning, memory, and attention impairments. From a clinical point of view, the first effect of cognitive impairments in drug-dependent patients has to do with problems in maintaining abstinence. For example, alterations in inhibitory control or decision-making processes delay an immediate reward for the benefit of their long-term recovery [282].

The cognitive decline associated with drug dependence is a disabling clinical condition because it significantly decreases the patient’s response to the treatment by increasing the dropout rates [283]. For this reason, in clinical practice, treatment planning should be adjusted to the neuropsychological assessment of these patients, taking into account any attentional biases, inhibitory difficulties or alterations in problem-solving, when choosing the type of psychological technique to apply. Preclinical studies offer advantages in terms of experimental control, enabling researchers to manipulate variables such as the type of drug, duration of exposure, and specific cognitive domains to be assessed. Despite the extensive literature in this area, the available information remains limited. A notable knowledge gap exists due to the absence of a clear correlation between the specific drug type, duration of consumption, timing of evaluation, and the resulting cognitive impairments. Addressing this gap is crucial and should be a priority for future research endeavors. At the clinical level, studying this topic poses challenges in identifying a homogeneous population for analysis, further complicating the identification of specific biomarkers associated with these clinical conditions. Consequently, the implementation of animal models that simulate relevant cognitive aspects following drug exposure holds significant value.

Animal studies can potentially enhance our understanding of the cognitive impact of drugs of abuse and the underlying molecular alterations involved. By providing a controlled environment, animal models can offer valuable insights that contribute to our knowledge in this field. Studies with animal models also provide valuable information on the potential brain regions and mechanisms involved in these cognitive alterations after exposure to a drug of abuse. Therefore, this review has also included behavioral and molecular information from animal studies. These studies support the results reported in humans on the strong involvement of different brain regions, such as the PFC and the HIPP, in cognitive impairment. On the other hand, dopaminergic, serotonergic, opioid, glutamatergic, and cannabinoid systems have also been described in clinical studies and animals. Interestingly, all the information from animal studies suggests that early exposure to drug misuse can induce worse cognitive performance in adulthood than exposure later in life. Adolescence is a particularly vulnerable stage characterized by higher neuronal and brain plasticity, easily affected by toxic substances.

In summary, this study provides valuable translational information on cognitive alterations in humans and rodents that may occur after acute or chronic drug misuse or abstinence exposure. In addition, the available information about the molecular changes has shown that more studies are required to further explore the drug of abuse-induced brain alterations involved in cognitive decline. In this line, different aspects should be considered, such as the stage and pattern of exposure, potential sex differences, and other comorbidities or risk factors. More studies are required to generate knowledge about these alterations in different, very specific stages of drug abuse. In addition, a better understanding of the molecular mechanisms involved in cognitive impairment in drug addiction may facilitate the identification of potential pharmacological targets for treating these alterations.

## Figures and Tables

**Figure 1 biomedicines-11-01796-f001:**
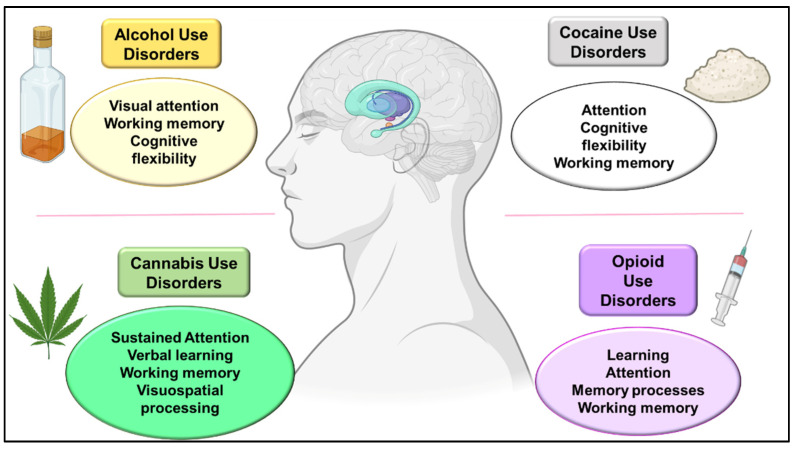
Main cognitive deficits observed in humans with alcohol use disorder, cannabis use disorder, cocaine use disorder, and opioid use disorder.

**Table 1 biomedicines-11-01796-t001:** Summary of the findings on alcohol-induced cognitive and biological changes in animal models.

ALCOHOL					
Drug and Pattern of Administration	Strain and Stage of Administration	Test	Behavioral Alterations	Biological Alterations	References
EtOH 2 g/kg, i.p., acute	Female Wistar albino rats, Adulthood	Spontaneous alternation in the Y maze	Reduced spontaneous alternation	-	[47]
EtOH 1.5 g/kg, i.p., acute	Male Sprague-Dawley rats, Adulthood	Eight-arm radial maze	Impaired contextual memory	-	[49]
EtOH 0.75, 1.5 and 2 g/kg, i.p., acute	Male Sprague-Dawley rats, Adulthood	Eight-arm radial maze	Impaired behavioral variability	-	[50]
EtOH 0.75 g/kg, i.p., acute	Male Long-Evans rats, Adulthood	Operant chambers	Working memory impairment	-	[48]
EtOH 3 g/kg, i.p., two administrations	Male Sprague-Dawley rats, Adulthood	-	-	↑ Doublecortin, astrogliosis and ↓ mature neurons in the SGZ of the DG ↓ Neuronal TLR4 in CA1 and DG	[51]
EtOH 2 g/kg, i.p., acute	Male C57BL/6J mice at 3, 5 and 7 weeks of age	Fear conditioning	No changes were observed	-	[53]
EtOH 1 g/kg, i.p., Acute, pre-training	Male NMRI mice, Adulthood	Step-down inhibitory avoidance	Reduction in aversive memory retention 24 h after the training session	Reduced pCREB/CREB protein ratio in the HIPP	[54]
EtOH 1.6 and 2.4 g/kg, i.p., Acute	Male C57BL/6J mice	Novel object recognition	Reduced object recognition	-	[56]
Chronic EtOH → EtOH 2% *v*/*v* to 20% *v*/*v* solution	Male Wistar rats, Adulthood	Novel object recognition Spatial recognition test	Impaired recognition in both tests after acute and chronic EtOH, with worse results in chronically treated rats	-	[57]
EtOH → 2 g/kg, i.p., once a week					
EtOH 5 g/kg, p.o., 2-day on/off cycle, 13 administrations	Male Sprague-Dawley rats Early Adolescence PND28 Male Sprague-Dawley rats Mid-adolescence PND35 Male Sprague-Dawley rats Adult PND65	Spontaneous alternation in a plus maze Non-spatial discrimination learning and reversal task	Impaired behavioral flexibility in the plus maze test Impaired acquisition and learning in the non-spatial discrimination test	Reduced BDNF protein levels in the PFC immediately after EtOH administration in all ages. In abstinence, BDNF reduction was maintained in early adolescents, whereas it increased in adults	[60]
EtOH 4 g/kg, p.o. daily for 11 days	Male and female Sprague-Dawley rats Early adolescence PND25 Male and female Sprague-Dawley rats Late adolescence PND45	Behavioral flexibility by operant set-shifting task	Males showed deficits in behavioral flexibility only after early ethanol exposure	-	[61]
EtOH vapor 14 h/day, 7–10 weeks	Male Wistar rats, Adulthood PND56	-	-	↓ Proliferation and differentiation of oligodendrocyte progenitor cells and ↓ Myelin expression in the mPFC	[62]
EtOH vapor (15–17 mg/L air), 16 h for 4 consecutive days	Male C57BL/6J mice, Adulthood	Reversal learning task Attentional set-shifting task	No changes in the reversal learning task ↑ Number of errors in the attentional set-shifting	↑ NMDA/AMPA ratio in the mPFC, with an increase of NMDA component	[63]
EtOH vapor 14 h/day, 15 days	Male Long-Evans rats, Adulthood	Operant set-shifting task	↑ Number of errors in the attentional set-shifting	↓ D2/D4 modulatory actions in the mPFC	[65]
EtOH vapor, 14 h/day, 4 cycles of 2-daily exposure	Male Long-Evans rats, Adolescence	Operant set-shifting task	↓ Behavioral flexibility	Positive allosteric modulation of mGluR5 reverses behavioral deficits	[64]
EtOH 3 g/kg, i.p., 8 intermittent administrations	Wistar rats, Adolescence	Conditional discrimination learning NOR	Cognitive deficits in adolescence and adulthood	↑ COX2 and iNOS levels ↑ cell death in the neocortex, HIPP and cerebellum	[68]
EtOH liquid diet over 4 weeks	Female C57BL/6J mice, Adulthood	NOR Y-maze	↓ Recognition of the novel object	↓ Neural stem cell proliferation and survival in the HIPP	[69]
EtOH 2–10%, 4 months, p.o.	Male Swiss mice, Late adolescence	NOR	↓ Recognition of the novel object	↑ γ protein levels	[70]
EtOH 5 g/kg (p.o.), intermittent administration for 16 days	Male Sprague-Dawley rats, Adolescence	-	-	↓ PSD-95 and SAP102	[71]
EtOH 5 g/kg (p.o.), intermittent administration for 20 days	Male Sprague-Dawley rats, Adolescence And Adults	-	-	↓ *I*_A_ mean peak amplitude	[72]
EtOH liquid diet with 7% alcohol, 1, 2 or 4 weeks (p.o.)	Male Sprague-Dawley rats,	-	-	↓ PCNA, DCX in the HIPP	[73]
EtOH 3–10% (p.o.), voluntary consumption	Male C57BL/6J mice, Adults	Fear conditioning test NOR Barnes maze test	↓ Behavioral flexibility, context-induced freezing and novel object recognition	↓ *BDNF* gene methylation and ↑ BDNF signaling pathway in CA1 and CA3 of the HIPP	[74]
EtOH 4 g/kg (p.o.) for 30 consecutive days	Male Wistar rats, Adults	Morris water maze	↓ spatial memory	↑ MDA, ↓ SOD and GPx in the HIPP	[75]
EtOH vapor 14 h/day, 7 weeks	Male Wistar rats, Adults	-	-	Spine density in the HIPP ↑ after EtOH exposure and ↓ in abstinence	[76]
EtOH 2 g/kg (p.o.) for 5 consecutive days	Male Sprague-Dawley rats, Adolescence	-	-	↑ NMDA receptor in the frontal cortex in the abstinence phase	[67]
Intermittent access to 20% of voluntary ethanol consumption followed by operant conditioning	Chat-eGFP, Char-Cre, VGlut2-Cre, Drd1a-tdTomato and Ai32 mice with C57BL/6J background	-	-	↓ Thalamic excitation of DMS CINs	[66]
EtOH 3 g/kg on 2 consecutive days with 48 h intervals over 2 weeks	Male and female C57BL6J mice, Adolescence	Hebb-Williams maze NOR Passive avoidance test	↓ Learning and memory in all paradigms	↓ P-CREB and P-ERK protein expression	[77]
EtOH intermittent exposure 5 g/kg, 2 days on and 2 days off	Male Wistar rats, Adolescence	NOR	↓ Recognition memory	↓ Axial and radial diffusivity in several brain regions	[78]
EtOH intermittent exposure 5 g/kg, 2 days on and 2 days off	Male and female rats	Spontaneous alternation NOR	No changes in adolescence, but a slight reduction in adult rats	Blunted behavioral-evoked HIPP acetylcholine efflux	[79]
EtOH intermittent exposure 5 g/kg (p.o.), 2 days on and 2 days off	Male and female Sprague-Dawley rats	Social recognition	↓ Social memory and learning	↓ Number of cholinergic interneurons in the NAc	[81]
Liquid EtOH diet 2.4–7.2%, p.o. for 20 days	Male Sprague-Dawley rats	Emotional novel object recognition	↓ Emotional learning	Loss of dendritic “long thin” spines in the NAc during abstinence	[82]
EtOH 4 g/kg, i.p. at 24 h of intervals for 5 days	Male Long-Evans rats, Adolescence and Adulthood	Morris water maze	↓ Spatial learning in adolescents but not in adults during the abstinence phase	-	[83]
EtOH 3.4 g/kg for 6 consecutive days	Male Wistar rats	-	-	↑ GLU release in the HIPP during the abstinence phase	[84]
54 cycles of liquid 10% EtOH diet followed by an abstinence period	Male Wistar rats	5-Choice serial reaction time task	↓ Behavioral inhibition and attentional capacity at 7 and 34 days of abstinence	-	[85]
EtOH 5 g/kg for 5 days + 6 g/kg for 55 additional days, followed by abstinence period	Male Sprague-Dawley rats	-	-	↓ Short and ↑ long-term HIPP CB1r protein levels	[86]
2 to 10% of EtOH for 5 months, followed by an abstinence period	Male C57BL/6WT and TLR4 knock-out mice	NOR	↓ Discrimination index	↑ Neuroinflammation damage	[87]

CB1r: cannabinoid receptor 1; CINs: cholinergic interneurons; DCX: doublecortin; DG: dentate gyrus; DMS: dorsomedial striatum; GLU: glutamate; GPx: glutathione peroxidase activity; HIPP: hippocampus; MDA: malondialdehyde; NAc: nucleus accumbens; NOR: novel object recognition; PCNA: proliferating cell nuclear antigen; PSD-95: post-synaptic density protein-95; SAP102: synapse-associated protein 102; IA: voltage-gated A-type potassium channel; SGZ: subgranular zone; SOD: superoxide dismutase activity; TLR4: toll-like receptor 4.

## Data Availability

Not applicable.

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
