# Peer review of "Cognitive Alterations in Addictive Disorders: A Translational Approach"

_biomedicines, 2023, doi:10.3390/biomedicines11071796_

Round 1
Reviewer 1 Report
In this well-written paper, the authors discussed the cognitive decline in drug abuse-addicted patients. Authors say that the most important changes are found during the dependence and drug abstinence phases but the associated cognitive decline may impair the response to addiction treatment and increase drop-out rates.
The authors in the paper continue their discussion by stating that the development of animal models to simulate drug abuse-induced learning and memory alterations could be useful to better clarify the related pathophysiological mechanisms.
The authors say also that the main goal of their paper was to summarize the principal recent findings of cognitive impairments and the associated biological markers in patients addicted to some of the most consumed drugs of abuse (alcohol, cannabis, cocaine, opioids and others as methamphetamine) and animal models simulating this clinical situation.
The paper is ambitious, quite long, and appears as a book chapter.
My main criticisms regard the complexity of the argument. In my opinion, the paper addresses too many aspects of the cognitive decline in drug abuse-addicted patients, discussing a few but omitting many.
- The section on alcohol is intriguing but is missing FASD.
- The other sections appear as a mere description of the arguments.
- The authors discuss the putative role of BDNF on the cognitive decline in drug abuse-addicted patients. However, NGF is totally missing and is missing too its role in the limbic cholinergic system. Why?
In conclusion, I do suggest the authors to better describe the exclusion/inclusion criteria of the studies’ selection, or alternatively, concentrate their efforts on one drug only.
Minor editing of the English language required
Author Response
Reviewer 1
In this well-written paper, the authors discussed the cognitive decline in drug abuse-addicted patients. Authors say that the most important changes are found during the dependence and drug abstinence phases but the associated cognitive decline may impair the response to addiction treatment and increase drop-out rates.
The authors in the paper continue their discussion by stating that the development of animal models to simulate drug abuse-induced learning and memory alterations could be useful to better clarify the related pathophysiological mechanisms.
The authors say also that the main goal of their paper was to summarize the principal recent findings of cognitive impairments and the associated biological markers in patients addicted to some of the most consumed drugs of abuse (alcohol, cannabis, cocaine, opioids and others as methamphetamine) and animal models simulating this clinical situation.
The paper is ambitious, quite long, and appears as a book chapter.
My main criticisms regard the complexity of the argument. In my opinion, the paper addresses too many aspects of the cognitive decline in drug abuse-addicted patients, discussing a few but omitting many.
The section on alcohol is intriguing but is missing FASD. The other sections appear as a mere description of the arguments. |
The main goal of the review was to summarize the principal findings about the cognitive impact of drug abuse. We agree with the reviewer regarding the importance of FASD in alcohol-related alterations. However, considering the complexity of these disorders and the very early stage of alcohol exposure, we believe that this new section could not be entirely in line with the subject of the manuscript and could be the objective of another work more focused on the perinatal exposure to different drugs of abuse. |
The authors discuss the putative role of BDNF on the cognitive decline in drug abuse-addicted patients. However, NGF is totally missing and is missing too its role in the limbic cholinergic system. Why? |
We decided to discuss the putative role of BDNF on cognitive impairment because it is the primary molecule involved in plastic changes related to learning and memory [1]. Although NGF is critical for the survival and maintenance of neurons and induces axonal growth, changes in BDNF expression are associated with aging and psychiatric disease in structures essential for memory processes, such as the hippocampus and parahippocampal areas [2]. However, in the last years, it was found that NGF metabolic pathway is impaired in Alzheimer's disease and other amyloid pathologies and could be considered a potential biomarker of cognitive decline in AD[3]. Following the reviewer's suggestion, we added the following paragraph in the 2.5.2 section: In the adult brain, BDNF preserves high expression levels and regulates excitatory and inhibitory synaptic transmission. High levels of BDNF have been detected in the hippocampus, amygdala, cerebellum, and cerebral cortex in humans, with the highest levels in hippocampal neurons. Nerve growth factor (NGF) is implicated in the maintenance of basal forebrain cholinergic neurons in adulthood. NGF metabolic pathway has been impaired in Alzheimer's disease and other amyloid pathologies. Although NGF is critical for the survival and maintenance of neurons and induces axonal growth, changes in BDNF expression are highest altered in structures essential for memory processes, such as the hippocampus and para-hippocampal areas. Therefore, our study focused on BDNF changes in substance use disorders associated with cognitive impairment. |
In conclusion, I do suggest the authors to better describe the exclusion/inclusion criteria of the studies' selection, or alternatively, concentrate their efforts on one drug only |
We thank the reviewer for this comment. We initially included a "Materials and methods" section describing the literature search procedure and the criteria used to select the articles. However, the editorial office asked us to remove this section. Following the reviewer's suggestion, we have now included the "Materials and Methods" section in the manuscript. Please, see page 2, line 82. |
- Liu, F., et al., Combined effect of nerve growth factor and brain‑derived neurotrophic factor on neuronal differentiation of neural stem cells and the potential molecular mechanisms. Mol Med Rep, 2014. 10(4): p. 1739-45.
- Miranda, M., et al., Brain-Derived Neurotrophic Factor: A Key Molecule for Memory in the Healthy and the Pathological Brain. Front Cell Neurosci, 2019. 13: p. 363.
- Cuello, A.C., R. Pentz, and H. Hall, The Brain NGF Metabolic Pathway in Health and in Alzheimer's Pathology. Front Neurosci, 2019. 13: p. 62.
Reviewer 2 Report
The Review article titled ‘Cognitive alterations in addictive disorders: a translational approach’ by Gasparyan A et al. summarizes cognitive impairment and associated biological markers in patients addicted to some of the most consumed drugs of abuse. Further, the review describes effects of drugs in animal models simulating various cognitive deficits.
The review is exhaustive and well written. It is a good description of cognitive deficits associated with drugs of abuse. The review can be useful to both clinicians and researches working in the field of drug addiction. The tables are good and summarize preclinical studies well. The authors can think of incorporating the following suggestions to improve the paper.
1. The information could be reorganized to include clinical deficit observed in humans followed by preclinical findings in animal model simulating the same deficit. For example, working memory- after describing working memory in patients- description of animal models assessing working memory with the same drug would be great. This would give the reader get a more comprehensive picture of the cognitive deficit and also give the reader an idea of the underlying molecular mechanism. If preclinical evidence for particular cognitive deficit for particular drug has not been demonstrated that would become more apparent with this kind of reorganization.
2. A major drawback of the article is that it primarily lists/reviews findings from published literature. However, the article has not identified gaps in knowledge.
3. Further the review does not describe future directions that may help move the field forward. The authors can incorporate this information from literature describing cognitive deficits not involving drugs of abuse.
4. Finally, the review does not identify pharmacological compounds that may have shown to reverse some of the cognitive deficits in preclinical studies that may still need to be evaluated in humans. Similarly, there may be some medications that have shown promise in clinical trials in humans with some cognitive deficits (with or without history of drug abuse) that could also be included in the review. This information could be included in a table or Box format.
5. A figure or table summarizing various cognitive deficits observed in humans abusing various drugs of abuse will help the readers quickly grasp the information
Author Response
Reviewer 2
The Review article titled ‘Cognitive alterations in addictive disorders: a translational approach’ by Gasparyan A et al. summarizes cognitive impairment and associated biological markers in patients addicted to some of the most consumed drugs of abuse. Further, the review describes effects of drugs in animal models simulating various cognitive deficits.
The review is exhaustive and well written. It is a good description of cognitive deficits associated with drugs of abuse. The review can be useful to both clinicians and researchers working in the field of drug addiction. The tables are good and summarize preclinical studies well. The authors can think of incorporating the following suggestions to improve the paper.
The information could be reorganized to include clinical deficit observed in humans followed by preclinical findings in animal model simulating the same deficit. For example, working memory- after describing working memory in patients- description of animal models assessing working memory with the same drug would be great. This would give the reader get a more comprehensive picture of the cognitive deficit and also give the reader an idea of the underlying molecular mechanism. If preclinical evidence for particular cognitive deficit for particular drug has not been demonstrated that would become more apparent with this kind of reorganization. |
We thank the reviewer for this observation. Although it would be interesting to organize the text according to the different cognitive disorders induced by each type of drug, from clinical to preclinical, we believe this would make the reading somewhat cumbersome. It should also be noted that the cognitive changes seen in clinical trials are not necessarily the same as those seen in animal studies, which could make integration very difficult. On the other hand, it is very likely that with this approach, there would be a significant imbalance between the different sections (cognitive disorders), with the possibility of having sections with results in patients but not in animals and vice versa. In short, we believe that the approach we have proposed is the one that best allows us to obtain a translational vision of the cognitive problems for each type of drug. |
A major drawback of the article is that it primarily lists/reviews findings from published literature. However, the article has not identified gaps in knowledge. |
The review has focused on the study of the effects of different drugs of abuse on cognitive performance. Indeed, despite the extensive literature in this field, the information available is scarce. The absence of a clear correlation between the type of drug, the duration of consumption and the time of evaluation with the induced cognitive alterations is a knowledge gap that should be addressed in future studies. At the clinical level, it is difficult to find a homogeneous population for the study, making identifying specific biomarkers of this clinical situation complex. These aspects point out the higher value of the implementation of animal models simulating some cognitive aspects after drug exposure. Preclinical studies allow better control of the experimental conditions, including the type of drug abuse, the exposure time, and the cognitive aspect of evaluation. Therefore, these types of studies could increase our understanding of the cognitive impact of drugs of abuse and the underlying molecular alterations. Please, see the additional information in the Discussion section. |
Further the review does not describe future directions that may help move the field forward. The authors can incorporate this information from literature describing cognitive deficits not involving drugs of abuse. |
As suggested by the reviewer, we added in the Discussion section potential future directions that should be addressed to increase the knowledge about cognitive impairment induced by drug abuse and the molecular mechanisms involved. |
Finally, the review does not identify pharmacological compounds that may have shown to reverse some of the cognitive deficits in preclinical studies that may still need to be evaluated in humans. Similarly, there may be some medications that have shown promise in clinical trials in humans with some cognitive deficits (with or without history of drug abuse) that could also be included in the review. This information could be included in a table or Box format. |
The main aim of this study was to provide a very comprehensive review of the main cognitive impairments induced by drugs of abuse in animals and humans. Although this topic suggested by the reviewer is also very relevant, it was not the focus of the work and could be the subject of another review manuscript. The manuscript is already 68 pages long and contains almost 300 references. |
A figure or table summarizing various cognitive deficits observed in humans abusing various drugs of abuse will help the readers quickly grasp the information |
We thank the reviewer’s suggestions. We agreed and decided to make a figure summarizing various cognitive deficits in substance use disorders (Figure 1). |
Round 2
Reviewer 1 Report
I do not agree with the authors' statement "Although NGF is critical for the survival and maintenance of neurons and induces axonal growth, changes in BDNF expression are highest altered in structures essential for memory processes, such as the hippocampus and para-hippocampal areas". Please remove the sentence on NGF.
Minor editing of the English language required
Author Response
Reviewer 1 #
I do not agree with the authors' statement "Although NGF is critical for the survival and maintenance of neurons and induces axonal growth, changes in BDNF expression are highest altered in structures essential for memory processes, such as the hippocampus and para-hippocampal areas". Please remove the sentence on NGF.
We have removed the sentence. Please, see the new version on lines 1217-1220.
In addition, the manuscript has been reviewed by a professional native translator and English mistakes have been corrected. Please, find attached the corresponding certificate.
